# Context parroting: A simple but tough-to-beat baseline for foundation models in scientific machine learning

**Yuanzhao Zhang**
Santa Fe Institute
Santa Fe, NM, USA
yzhang@santafe.edu

**William Gilpin**
University of Texas at Austin
Austin, TX, USA
gilpin@chaos.utexas.edu

## Abstract

Recent time-series foundation models exhibit strong abilities to predict physical systems. These abilities include zero-shot forecasting, in which a model forecasts future states of a system given only a short trajectory as context, without knowledge of the underlying physics. Here, we show that foundation models often forecast through a simple parroting strategy, and when they are not parroting they exhibit some shared failure modes such as converging to the mean. As a result, a naive context parroting model that copies directly from the context scores higher than leading time-series foundation models on predicting a diverse range of dynamical systems, including low-dimensional chaos, turbulence, coupled oscillators, and electrocardiograms, at a tiny fraction of the computational cost. We draw a parallel between context parroting and induction heads, which explains recent works showing that large language models can often be repurposed for time series forecasting. Our dynamical systems perspective also ties the scaling between forecast accuracy and context length to the fractal dimension of the underlying chaotic attractor, providing insight into previously observed in-context neural scaling laws. By revealing the performance gaps and failure modes of current time-series foundation models, context parroting can guide the design of future foundation models and help identify in-context learning strategies beyond parroting.

## 1 Introduction

A key test of generalization in scientific machine learning (SciML) is zero-shot forecasting: the ability to forecast future states of a new physical system based on a short context trajectory. Prior SciML approaches primarily focus on developing specialized forecasting models trained specifically on the system that needs to be predicted (Brunton et al., 2016; Weinan, 2017; Chen et al., 2018; Pathak et al., 2018; Li et al., 2020; Chen & Tao, 2021; Gauthier et al., 2021; Lim & Zohren, 2021; Karniadakis et al., 2021; Levine & Stuart, 2022; Mikhaeil et al., 2022; Brunton et al., 2022; Das et al., 2023; Krishnapriyan et al., 2023; Yang et al., 2024; Yu & Wang, 2024; Azizzadenesheli et al., 2024; Brenner et al., 2024a;b; Ricci et al., 2024; He et al., 2025; Cheng et al., 2025; Grigoryeva et al., 2025; Berry & Das, 2025). However, the generality of these models is limited by the amount of system-specific data available, motivating the recent development of time-series foundation models (Oreshkin et al., 2021; Garza & Mergenthaler-Canseco, 2023; Rasul et al., 2023; Jin et al., 2023; Zhou et al., 2023; Gruver et al., 2024; Dooley et al., 2024; Liu et al., 2024b; Woo et al., 2024; Ansari et al., 2024; Goswami et al., 2024; Das et al., 2024; Liang et al., 2024; Shi et al., 2025; Zhai et al., 2024; Liu et al., 2025b), which are trained on vast amounts of observed and simulated time series from diverse domains, and which can subsequently perform zero-shot forecasts for any time series—including those generated by previously-unseen dynamical systems. Interestingly, it was recently found that, when available historical data is limited, time-series foundation models outperform classical deep learning models in forecasting chaotic dynamical systems (Zhang & Gilpin, 2024).

What mechanisms do time-series foundation models use to make zero-shot forecasts, and why they are effective for dynamical systems not seen during pre-training? It was recently observed that one such foundation model, Chronos (Ansari et al., 2024), often employs an extremely simple strategy when

forecasting chaotic systems (Zhang & Gilpin, 2024). The strategy, *context parroting*, scans the context for nearly repeating motifs and copies the part of the context following the best-matching motif as its prediction (Fig. 1). This can be viewed as a kind of "in-context nearest neighbor" algorithm, which is easy to implement during in-context computation (Garg et al., 2022). How good is context parroting as a zero-shot forecasting strategy? By comparing it with existing foundation models, what can we learn about current models' strengths and limitations?

Here, we compare context parroting with a diverse set of competitive baselines on the challenging task of forecasting chaotic systems. Our baselines include four state-of-the-art time-series foundation models: Chronos and Chronos Bolt (Ansari et al., 2024), TimesFM (Das et al., 2024), Time-MoE (Shi et al., 2025), and Moirai (Woo et al., 2024), as well as a recent foundation model specifically designed for dynamical systems: DynaMix (Hemmer & Durstewitz, 2025). In the Appendix, we also include two classical forecasting methods that are particularly effective in the small-data limit: AutoARIMA (Hyndman & Athanasopoulos, 2018) and simplex projection (Sugihara & May, 1990). The latter represents a classical nonlinear forecasting method conceptually resembling context parroting (Appendix F). We find that parroting outperforms all baselines (including the leading foundation models) in both zero-shot forecast accuracy and inference cost, especially for longer context windows. Our results suggest that current time-series foundation models do not fully utilize the information in the context data, and thus still have significant room for improvement when it comes to SciML tasks.

Our main contributions are:

1. Introduce context parroting as a simple but effective baseline for zero-shot forecasting of dynamical systems, which can guide the design of more informative benchmarks that cannot be solved by simple repetitions and help identify forecasting strategies beyond parroting

2. Show that context parroting outperforms leading time-series foundation models in predicting chaotic systems and reveal common failure modes of many existing foundation models, which can guide the design of better models in the future

3. Explain the in-context neural scaling law between forecast accuracy and context length, linking the scaling coefficient to the fractal dimension of the underlying chaotic attractor

## 2 RELATED WORK

**Foundation models for science.** Foundation models have recently been introduced for many scientific machine-learning tasks (Miller et al., 2024), including partial differential equations (Takamoto et al., 2022; Yang et al., 2023; Rahman et al., 2024; Subramanian et al., 2024; Herde et al., 2024; McCabe et al., 2024; Totounferoush et al., 2025), neuroscience (Cui et al., 2024; Caro et al., 2023; McKeen et al., 2024), and weather forecasting (Nguyen et al., 2023; Bodnar et al., 2024). However, most of these foundation models remain a black box, and they have not yet provided interpretable strategies for forecasting diverse physical and dynamical processes. Here, we analyze context parroting as a simple mechanism used by time-series foundation models, noting its strengths and weaknesses as a zero-shot forecasting strategy. This strategy, and the insights gained here, can potentially be applied to other scientific tasks.

**In-context neural scaling laws.** Neural scaling laws describe the relationship between the performance of a neural network and certain resources, such as model size, data size, or the amount of compute (Kaplan et al., 2020; Sorscher et al., 2022; Bahri et al., 2024; Yao et al., 2024). Such scaling laws allow practitioners to predict the performance of yet-to-be-trained models based on the available resources and allocate them strategically to optimize compute-adjusted accuracy (Hoffmann et al., 2022). When applying LLMs to forecast dynamical systems, Liu et al. (2024a) recently observed an in-context neural scaling law, in which the test loss decreases with the context length following a power law. Here, we show that this in-context neural scaling law can be reproduced when using context parroting to predict dynamical systems, and the scaling coefficient can be linked to an invariant property of the underlying dynamic process (the fractal dimension of the chaotic attractor). This finding shows that neural scaling laws are intrinsically linked to invariant properties of the process generating the data, and the theory can potentially be generalized to other models and tasks. For example, can we estimate the "fractal dimension" of a language from the neural scaling laws of LLMs (Du & Tanaka-Ishii, 2025)?

**In-context learning and induction heads.** Induction heads are computational circuits that naturally emerge in simple transformers through training, and they have been hypothesized to underlie much of the in-context learning ability of foundation models (Elhage et al., 2021; Olsson et al., 2022; Von Oswald et al., 2023; Reddy, 2023). In its simplest form, an induction head copies repeating tokens in the context to make predictions. For example, when presented with a token stream $[A][B] \dots [A]$, an induction head will output $[B]$ as the next token. Prior works train transformers on minimal Markov chain grammars, and find that, during pretraining, models learn to identify increasingly higher-order $k$-grams, with different attention heads specializing in copying, lookup, and aggregation (Edelman et al., 2024; Chen et al., 2024a). These works imply that pretraining enables models to learn conditional distributions, allowing them to represent sequence distributions seen in the context (Lv et al., 2024; Chen et al., 2024b; Keskar et al., 2019; Zekri et al., 2024).

There is a clear parallel between context parroting and induction heads: both are essentially copy-and-paste operations, with context parroting involving the matching of not just one but multiple contiguous tokens. In fact, it is easy to imagine context parroting emerging naturally from combining multiple induction heads. This parallel can potentially explain the unreasonable effectiveness of applying language models trained on text to time-series tasks without fine-tuning or prompt engineering (Garza & Mergenthaler-Canseco, 2023; Jin et al., 2023; Zhou et al., 2023; Gruver et al., 2024; Liu et al., 2024a). The induction heads formed from training on natural language happen to be also effective for predicting time series and can be easily repurposed to implement strategies such as context parroting.

## 3 CONTEXT PARROTING AS A ZERO-SHOT FORECASTING STRATEGY

**Overview of context parroting.** In this section we motivate and introduce our baseline: context parroting. It was inspired by recent observations that Chronos often predicts chaotic systems by copying directly from the context (Zhang & Gilpin, 2024). An example of Chronos using parroting to forecast a partially-observed Lorenz system is shown in Fig. 1.

On a high level, context parroting uses the last $D$ tokens of the context to query the remaining context. For whatever context sequence that most closely matches the query, the subsequent tokens in the context are copied and used as the forecast. Because the length of the motif $D$ can be seen as the number of delayed states in a delay embedding from the lens of Takens' embedding theorem (Takens, 2006), we also refer to $D$ as the embedding dimension and will use the terms embedding dimension and query length interchangeably. Interpreting $D$ as the embedding dimension is convenient because context parroting can be seen as a nearest neighbor algorithm in the $D$-dimensional delay-embedded space. During the matching process, we exclude the last $D$ motifs to avoid parroting too close to where the prediction starts. Framed in terms of induction heads, the query lookup acts as a *copy* head, the nearest-neighbor match is a *selector*, and the exact repetition is the *aggregation* operation (Chen et al., 2024a). We provide a pseudocode for context parroting in Algorithm 1.

---

**Algorithm 1** Context Parroting

**Input:** Context trajectory $x_{1:L} = \{x_1, \dots, x_L\}$, embedding dimension $D$ (i.e., the length of the motif to match), and forecast length $H$.
**Output:** Forecast trajectory $x_{L+1:L+H} = \{x_{L+1}, \dots, x_{L+H}\}$.
  1: **for all** length-$D$ motif $s$: $x_{s-D+1:s}$ in the context $x_{1:L-D}$ **do**
  2:     compute the Euclidean distance $d_s$ between motif $s$ and the last motif $x_{L-D+1:L}$
  3: Find the best-matching motif, $s_{opt}$, with the smallest Euclidean distance
  4: Set the first $L - s_{opt}$ predicted points to be $x_{L+1:2L-s_{opt}} = x_{s_{opt}+1:L}$ and repeat until the forecast length $H$ is reached

---

**Relationship to classical nonlinear forecasting methods**. We show in Appendix F.3 that, in various limits, context parroting is equivalent to two classical algorithms from nonlinear dynamics: the *simplex projection* technique and the *S-map* algorithm (Sugihara & May, 1990; Sugihara, 1994). Both approaches have their foundations in Takens' embedding theorem, which states that time-delayed low-dimensional observables derived from a nonlinear dynamical system can recover key geometric properties of the underlying high-dimensional attractor (Takens, 2006). However, unlike context parroting, which looks for the best matching motif, simplex projection tries to identify multiple

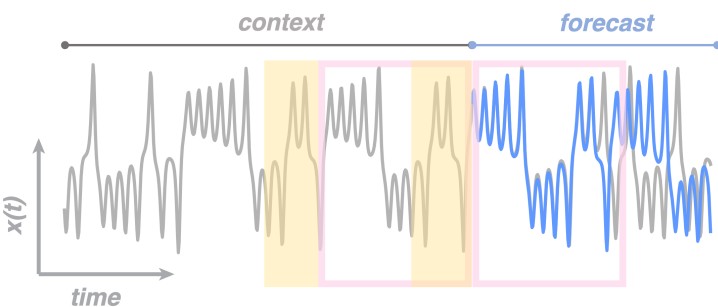

Figure 1: **Example of a foundation model forecasting chaotic dynamics with context parroting**. Here, the foundation model (Chronos) was asked to predict the $x$ variable of the Lorenz system based on a short context trajectory with 512 data points. Blue is the prediction and gray is the ground truth. Chronos produced an accurate prediction by simply looking for a motif in the context similar to the motif immediately preceding the prediction (highlighted in yellow) and copying the evolution following the matching motif (highlighted by pink boxes). We distill this context parroting strategy into Algorithm 1 and compare it against time-series foundation models (including Chronos itself).

matching motifs in the context and computes a weighted average as its forecast. This can potentially make simplex projection more sensitive to the choice of the embedding dimension $D$, limiting the method to small embedding dimensions in practice (Chang et al., 2017). Other than simplex projection and S-map, there also exists other zero-shot forecasting strategies from nonlinear dynamics. For example, the Farmer-Sidorowich method (Farmer & Sidorowich, 1987) looks at multiple nearest neighbors in the context and builds a local linear model to make forecasts. An interesting future direction is to compare these methods from nonlinear dynamics with context parroting and time-series foundation models, which might inspire new zero-shot forecasting strategies.

## 4 METHODS

**Datasets.** The `dysts` dataset provides a standardized benchmark of 135 low-dimensional chaotic systems, each defined by a set of ordinary differential equations between dimensionality three and six (Gilpin, 2021). The chaotic systems are drawn from different published papers and span fields such as neuroscience, climate science, fluid dynamics, and astrophysics. Every system is annotated with its largest Lyapunov exponent $\lambda$, an invariant characteristic of the underlying dynamics that quantifies the rate at which small perturbations grow over time. In chaotic systems, even minor errors rapidly compound over a characteristic timescale known as the Lyapunov time, defined as $\tau \equiv \lambda^{-1}$. To normalize the difficulty of predicting different chaotic systems (so results from the 135 systems can be meaningfully compared and combined), we generate trajectory data with a fixed sampling rate of 30 points per Lyapunov time, and also measure the forecast performance in terms of Lyapunov times. To show the relevance of our findings to a broad class of SciML tasks, later we also go beyond low-dimensional chaotic systems and simulated data by benchmarking on real-world datasets from ECG measurements and electronic circuits.

**Models.** For time-series foundation models, we select Chronos$_{\text{base}}$ (200M parameters), its variant Chronos-Bolt$_{\text{base}}$ (205M parameters), Time-MoE$_{\text{large}}$ (200M parameters), TimesFM-2.0 (500M parameters) and Moirai-2.0$_{\text{small}}$ (11M parameters) (Das et al., 2024; Ansari et al., 2024; 2025; Shi et al., 2025; Liu et al., 2025a). All of these models are pretrained on massive amounts of real-world time series data (hundreds of billions of data points), which are often complemented by synthetic data to improve generalization. We also consider DynaMix, a foundation model pretrained on chaotic dynamical systems (Hemmer & Durstewitz, 2025). These models encompass a wide array of design choices: Time-MoE, TimesFM-2.0, and Moirai-2.0 are decoder-only architectures, Chronos is an encoder-decoder architecture, and DynaMix is an almost-linear RNN trained via teacher forcing (Brenner et al., 2024a). Chronos and Time-MoE use pointwise tokenization, DynaMix implicitly tokenizes pointwise, while TimesFM-2.0 and Moirai-2.0 use patching. Time-MoE, DynaMix, and TimesFM-2.0 by default give point forecasts, whereas Chronos, Chronos-Bolt, and Moirai-2.0 provide probabilistic forecasts with uncertainty quantification. For these models, we use the median prediction when evaluating forecast errors. An important parameter for all foundation models is the

maximum context length $L_{\max}$, which varies from 512 data points (Chronos), 1680 (Moirai-2.0), 2048 (TimesFM-2.0), 4096 (Time-MoE), to arbitrary for DynaMix due to its recurrent formulation.

**Pipelines.** To evaluate different models' ability to zero-shot forecast dynamical systems, we generate a chaotic trajectory of length $10^5$ for each of the 135 chaotic systems in `dysts`, with a granularity of 30 data points per Lyapunov time. Each trajectory is normalized to have zero mean and unit standard deviation. For a given context length $L$, we randomly pick a length-$L$ segment from the chaotic trajectory and provide it to the model as the context. The model's task is to predict the next 300 data points (equivalent to 10 Lyapunov times) solely based on the context. We ask the model to make a univariate forecast on each dimension independently, which is then evaluated separately for each dimension. To obtain reliable statistics, we aggregate the results over all 135 chaotic systems, all dimensions, and 20 random initial conditions for each system.

**Metrics.** In line with previous research (Hyndman & Koehler, 2006; Makridakis et al., 2022; Gilpin, 2021; 2023), we assess forecasting performance using a diverse set of complementary metrics.

*Symmetric Mean Absolute Percentage Error (sMAPE).*

$$\text{sMAPE}(\mathbf{x}, \hat{\mathbf{x}}) \equiv 2\frac{100}{T}\sum_{t=1}^{T}\frac{|\mathbf{x}_t - \hat{\mathbf{x}}_t|}{|\mathbf{x}_t| + |\hat{\mathbf{x}}_t|},$$

where the sequence $\mathbf{x}_1, \mathbf{x}_2, \ldots, \mathbf{x}_T$ denotes the ground truth, and $\hat{\mathbf{x}}_1, \hat{\mathbf{x}}_2, \ldots, \hat{\mathbf{x}}_T$ are the corresponding predictions made by the model. To provide some context to help interpret the sMAPE value, we note that predicting the mean of white noise would give you an sMAPE around 200.

Other than sMAPE, we also show benchmark results using *Mean Square Error* (MSE) and *Mean Absolute Error* (MAE), two other metrics commonly used in the time series literature.

For chaotic dynamical systems, point forecasts will inevitably fail due to the exponential rate of error accumulation. It is thus equally important for a forecasting model to preserve the long-term invariant properties of the chaotic dynamics, such as Lyapunov exponents and the attractor dimension. Here, we compare the structure of true and predicted attractors by calculating the KL Divergence between their distributions.

*Kullback–Leibler Divergence between Attractors ($D_{\text{stsp}}$).*

$$D_{\text{stsp}} \equiv D_{\text{KL}}(P\|Q) = \sum_{x\in\mathcal{X}} P(x)\log\frac{P(x)}{Q(x)},$$

where $P$ and $Q$ represent the true and the predicted attractor, respectively. When estimating $D_{\text{stsp}}$, we follow the methodology in Hess et al. (2023); Göring et al. (2024). Specifically, we place Gaussian kernels at each point in the true and predicted trajectories and estimate the KL divergence between these Gaussian mixtures using a sampling-based approximation (Hershey & Olsen, 2007).

In addition, we analyze the frequency content of the predicted trajectories by computing the *Power Spectrum*. This is estimated using Welch's method. We quantify the accuracy of the spectral reconstruction by calculating the Hellinger distance between the true and predicted power spectra.

In the appendix, we also measure attractor reconstruction accuracy by comparing *Correlation Dimension* and the rate of divergence using *Lyapunov Exponents*. The correlation dimension estimates the fractal attractor dimension from a time series by calculating the scaling of the number of attractor points that fall within a given radius of each point (Grassberger & Procaccia, 1983).

## 5 RESULTS

### 5.1 CONTEXT PARROTING OUTPERFORMS FOUNDATION MODELS IN FORECASTING CHAOS

Here, we compare context parroting and foundation models in their ability to predict chaotic dynamics. Figure 2 shows each model's forecasting error (measured by sMAPE) as well as their accuracy in attractor reconstruction (measured by KL Divergence). It is clear that context parroting is better than all foundation models tested here in both metrics. In Appendix Fig. 7, we show that this remains true when benchmarked against MSE and MAE. The results for fractal dimension accuracy are shown in

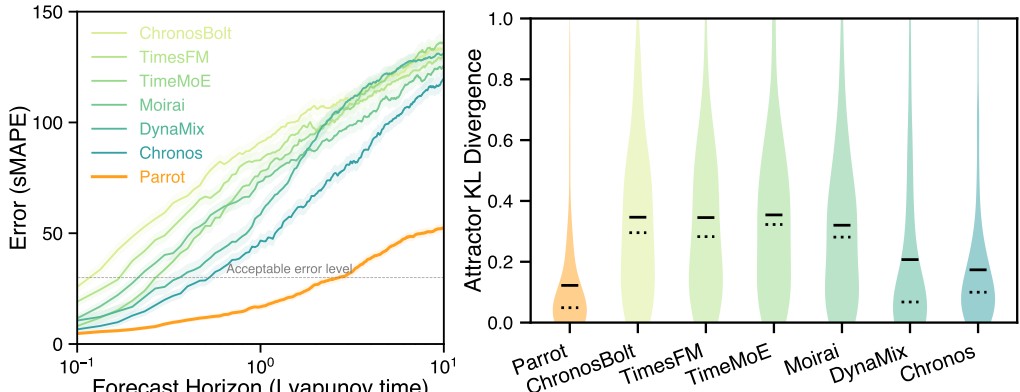

Figure 2: **Context parroting outperforms foundation models in zero-shot forecasting for both short-term point-wise accuracy and long-term attractor reconstruction**. Left: Forecast error of each model as a function of the forecast horizon. The context length is set to 512 for all models. Right: KL Divergence between the predicted attractors and the true attractors (smaller is better). Solid lines represent mean and dotted lines represent median. All results are obtained from 135 chaotic systems in the `dysts` database, with 20 trajectories from random initial conditions for each system.

Appendix Fig. 8, where parroting, DynaMix, and Chronos significantly outperform other foundation models.

Consistent with Hemmer & Durstewitz (2025), we observe that DynaMix preserves long-term geometry—termed the "climate" in classical forecasting (Lu et al., 2018). We attribute this capability to its recurrent architecture, which often better sustains oscillations over long horizons (Elman, 1990; Durstewitz, 2017). We also note that the efficient recurrent mixture-of-experts design of DynaMix allows it achieve competitive performance with a small fraction of the active parameters of other models.

Among transformer-based models, Chronos is the best performer in predicting chaotic systems, which is not surprising given that it often utilizes parroting as a key forecasting strategy on this dataset (Zhang & Gilpin, 2024). Chronos's tendency to context parrot arises from its distinct architecture as a language model operating on quantized time series. As a result, Chronos is trained using cross-entropy loss, which incentivizes preservation of k-gram frequencies and encourages the generation of diverse forecast samples consistent with the dynamical system's underlying measure (Yu et al., 2025). In contrast, TimeMoE and TimesFM are trained using mean squared error loss. These models lose diversity and regress to the mean at long forecast horizons, thus suppressing oscillations. Representative forecasts from the foundation models are shown in Appendix Fig. 6, which demonstrate that regressing to the mean is a common failure mode for many foundation models when forecasting chaotic systems.

Moreover, the inference cost of context parroting is negligible compared to transformer-based foundation models, not to mention the substantial GPU time needed to pretrain them. For example, a six orders of magnitude computational gap separates Chronos and context parroting for all context lengths. Combined with the fact that the performance of parroting is not sensitive to the choice of the embedding dimension $D$ (Appendix Fig. 9), these results establish context parroting as a simple but effective baseline for zero-shot forecasting of dynamical systems.

Given that context parroting by definition outputs a periodic prediction, one might suspect that it could fail to truly capture key invariant properties crucial to chaos. For example, the cyclic patterns produced by parroting should have its largest Lyapunov exponent equal to zero. However, parroted trajectories can have positive *finite-time* Lyapunov exponents, which is the only thing one can practically estimate from finite-length time series. Since parroting can in principle copy an arbitrarily long trajectory, the estimated Lyapunov exponents will approach the ground truth for long enough context. In this sense, parroting is able to capture invariant dynamical properties of chaos such as Lyapunov spectra and power spectra, despite only producing periodic trajectories by definition.

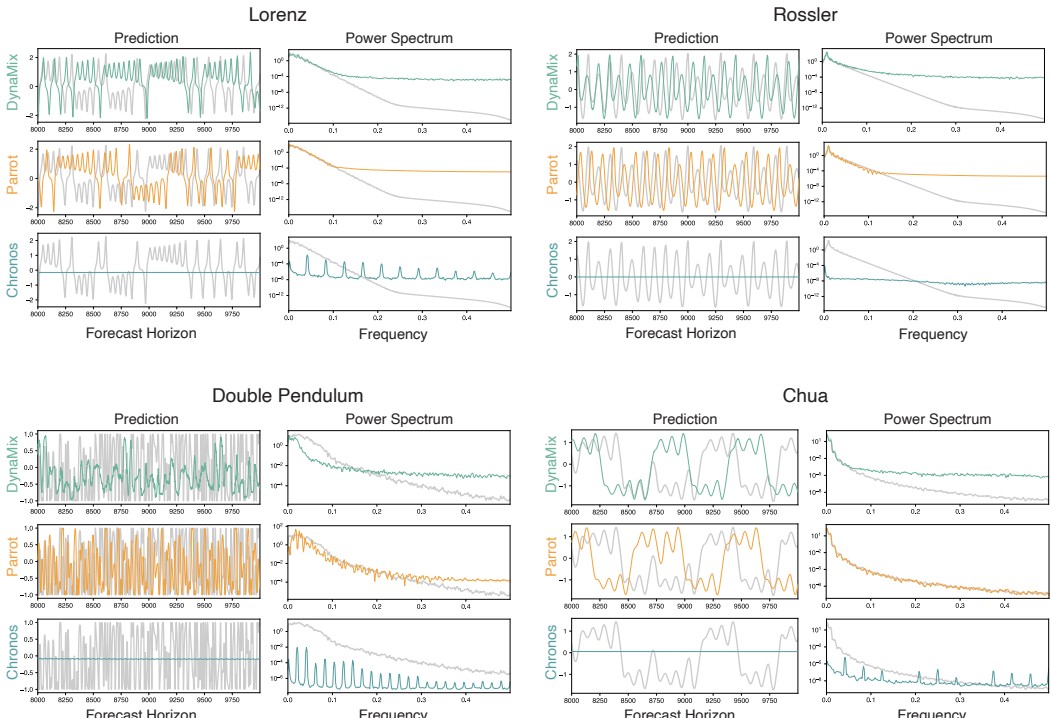

Figure 3: **Context parroting best reconstructs the power spectra of chaotic systems despite its predictions being periodic.** Predictions of four distinct chaotic systems using the two best-performing models DynaMix and Chronos, relative to Parroting. We set the context length to $L = 2000$. Forecasts are generated for $10,000$ points past the end of the context, and the last 2000 timepoints are shown. The power spectrum is estimated using Welch's method on the last 5000 timepoints.

In Fig. 3, we show that parroting can better capture the power spectrum than Chronos and DynaMix, two foundation models that achieved the lowest attractor KL divergence according to Fig. 2. We demonstrate the same for other invariant properties in Appendix Table 4 and Fig. 11. Moreover, we note that as the context length becomes longer, context parroting will be able to parrot increasingly complex patterns with longer periods. This is shown explicitly in Appendix Fig. 12, where parroting captures the power spectrum increasingly better as context length is increased.

Figure 4 further explores the effect of context length on forecast accuracy. We find that longer context windows produce better performance for context parroting, DynaMix, and Chronos. However, Chronos can effectively utilize a context length of at most $L = 512$, a limitation addressed in later generations models (Ansari et al., 2025). The need to set maximum context length before pretraining is a limitation of transformer-based architectures. In contrast, recurrent models like DynaMix, state-space models, and context parroting (Gu et al., 2021), do not share this limitation, allowing them to scale to contexts of arbitrary length during inference.

Interestingly, Chronos outperforms context parroting on short contexts, indicating that it employs additional zero-shot learning strategies beyond parroting. We expect that, at short context length, the time series becomes effectively nonstationary, which is the strength of time-series foundation models. For example, Chronos is great at continuing the local trend in the context, which can be a more effective strategy than parroting when the length of the context is limited.

Moreover, even when restricted to parroting, the $\sim \mathcal{O}(L^2)$ operations performed by attention heads in transformers like Chronos have, in principle, sufficient computational complexity to dynamically choose the optimal embedding dimension $D$ for each individual time series, giving attention an advantage over parroting algorithms with a fixed $D$, which have the $\sim \mathcal{O}(D\,L)$ complexity of nearest-neighbor search. It would be interesting to explicitly identify the mechanisms that enable Chronos to outperform parroting in the short-context regime.

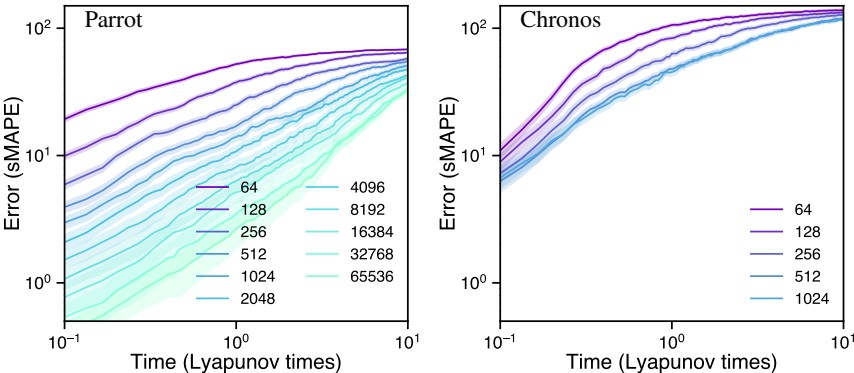

Figure 4: **Parroting can better utilize longer context data while Chronos does better for shorter contexts**. Each line represents the forecast error for a different context length. The performance of Chronos saturates once the context length exceeds its designed upper limit of $512$ data points, whereas the accuracy of context parroting keeps improving for longer context windows. Here we set the embedding dimension $D = 5$ for the parroting algorithm.

## 5.2 THEORETICAL EXPLANATION OF IN-CONTEXT NEURAL SCALING LAWS

Due to its simplicity and intimate connections to foundation models, context parroting can serve as an analytically-tractable model to explain some intriguing empirical findings from the literature. Recently, Liu et al. (2024a) reported an in-context neural scaling law for LLMs applied to dynamical systems, in which the one-step forecast error decreases algebraically with context length. However, it is unclear where this scaling law came from or why LLMs trained on text can be effective for time series without fine tuning. Here, we show that context parroting naturally gives rise to the same in-context scaling law and provides geometric insights into its origin. Given the similarity between parroting and the induction heads implemented by LLMs (Olsson et al., 2022), the geometric explanation we develop next for parroting not only applies to time-series foundation models, but can also conceivably be adapted to LLMs and partially explain the observations in Liu et al. (2024a).

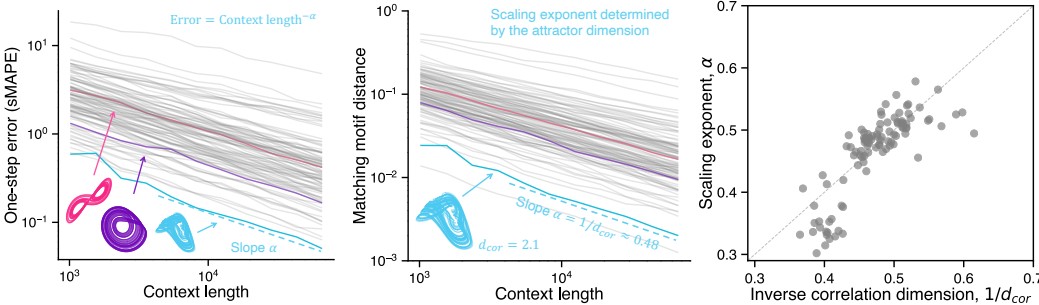

Figure 5: **Context parroting explains in-context neural scaling laws and links scaling exponents to attractor dimensions.** Left: One-step forecast error versus context length. Each line represents a distinct chaotic system and the scaling follows a power law. In principle, infinite context length should give parroting infinite accuracy for any deterministic system with a well-defined attractor. Middle: Euclidean distance between the last context point and its nearest neighbor in the delay-embedding space, as a function of context length. Again, the scaling follows a power law with the same scaling exponent $\alpha$. The scaling exponent can be different for each system and is linked to the correlation dimension $d_{\mathrm{cor}}$ of the chaotic attractor through the relation $\alpha = 1/d_{\mathrm{cor}}$. Right: Estimated scaling exponent $\alpha$ versus the inverse correlation dimension $1/d_{\mathrm{cor}}$, where each dot represents a chaotic system. For all panels we set the embedding dimension $D = 10$ for the parroting algorithm.

The left panel in Fig. 5 shows the power law relation between one-step forecast error (measured by sMAPE) and context length for parroting. Longer context lengths improve predictions by allowing the algorithm to find better matching motifs, and thus shadow the ground truth for longer. The middle panel in Fig. 5 shows improving overlap, with the distance between the matching motifs

decreasing algebraically with context length. The matching motif distance $\ell$ maps linearly to the one-step forecast error $e$ on average: $\langle e \rangle = c \langle \ell \rangle$, where $c$ is a system-specific constant that depends on the largest Lyapunov exponent of the chaotic dynamics. As a result, the power law for the matching motif distance implies the power law for one-step forecast error. Moreover, the two power laws share the same scaling exponent: $e \propto L^{-\alpha}$ and $\ell \propto L^{-\alpha}$, as confirmed by Fig. 5.

Can we predict the scaling coefficient $\alpha$ (how fast the forecast improves with more context data) for any given chaotic system? We link $\alpha$ to the geometric invariants of the underlying dynamics, specifically the fractal dimension of the chaotic attractor, as estimated by the correlation dimension:

$$d_{\text{cor}} \equiv \lim_{\epsilon, \epsilon' \to 0^+} \frac{\ln\left[\frac{C(\epsilon)}{C(\epsilon')}\right]}{\ln\left(\frac{\epsilon}{\epsilon'}\right)},$$

where $C(\epsilon)$ is the number of point pairs in the attractor that are within a given radius $\epsilon$ from each other. In other words, if we plot $C(\epsilon)$ against $\epsilon$ on a log-log plot, $d_{\text{cor}}$ would be the slope of the plotted line. We assume the chaotic systems are ergodic, so the a sufficiently-long context trajectory represents a random sample of the attractor. Longer context trajectories contain more samples and, for large enough sample sizes, we can show from the definition of the correlation dimension that the expected distance between a sample point and its nearest neighbor decreases with context length (i.e., the sample size) as $L^{-1/d_{\text{cor}}}$. For example, for a two-dimensional attractor, the distance between a random point and its nearest neighbor will decrease as $1/\sqrt{L}$. Fractal dimension thus measures the speed at which the distance between neighboring points on an attractor can be reduced by including more samples, and higher dimensionality requires more points to reduce the distance to the same extent. Mathematically, we thus expect $\alpha = 1/d_{\text{cor}}$. A similar scaling law has been derived for the Farmer-Sidorowich forecasting method from the nonlinear dynamics community (Farmer & Sidorowich, 1987). Indeed, in the right panel of Fig. 5, we observe strong correlation between $d_{\text{cor}}$ and $1/\alpha$ (Spearman correlation around $0.85$), supporting our theoretical argument above.

We show in Appendix Fig. 10 that the power law scaling persists when aggregated over chaotic systems, across a range of embedding dimension $D$. Although we focus on deterministic ODEs in Fig. 5, we note that the same power law scaling is expected to hold for discrete maps and weakly-stochastic systems. Explaining the power law for strongly-stochastic systems, such as random Markov chains (Liu et al., 2024a), is a promising direction for future research.

### 5.3 SciML tasks beyond low-dimensional chaotic systems

So far we focused on low-dimensional chaotic systems from the `dysts` dataset, which enabled systematic comparison between different forecasting models with standardized benchmarks. Here, we show that parroting also outperforms foundation models on a broader class of SciML tasks, including real-world datasets of current scientific interest. Our datasets are: (1) the von Karman vortex street at Reynolds number $\text{Re} = 900$, a standard problem in fluid dynamics representing a flow exhibiting intermittency. We generated time series corresponding to the top PCA modes, in order to capture global structure; (2) electrocardiogram recordings (via the QT Database in PhysioNet); (3) 28 coupled electronic circuits measured experimentally from Vera-Ávila et al. (2020)); and (4) 23 Kuramoto oscillators coupled through frustrated and nonreciprocal interactions, recently studied in León & Pazó (2025). These are all high-dimensional systems, two generated from simulations and two measured in the real world. For the metrics, we use MAE and MSE to measure pointwise forecast accuracy, and KL Divergence to measure the accuracy in attractor reconstruction. The results are summarized below. Parroting is the only model that ranks in the top three for all tasks and all metrics. Other metrics, such as valid prediction time and fractal dimension accuracy, give similar rankings.

Table 1: Performance comparison (**MAE @ 50 steps**, mean $\pm$ standard deviation) of forecasting models across SciML tasks. **Bold = best**, *italic = second and third best*.

| Task | Parrot | DynaMix | Chronos | Chronos Bolt | TimesFM | TimeMoE | Moirai |
|---|---|---|---|---|---|---|---|
| Turbulence | *0.403±0.210* | 0.505±0.247 | 0.431±0.237 | 0.567±0.247 | 0.510±0.174 | *0.394±0.172* | **0.382±0.189** |
| ECG | **0.624±0.315** | 0.777±0.241 | 0.873±0.422 | 0.752±0.279 | *0.723±0.259* | 0.799±0.158 | *0.684±0.237* |
| Circuit | **0.083±0.050** | 0.425±0.172 | *0.111±0.065* | 0.349±0.120 | *0.196±0.090* | 0.206±0.102 | 0.213±0.093 |
| Kuramoto | **0.004±0.001** | 0.076±0.002 | 0.072±0.029 | 0.961±0.084 | 0.624±0.061 | *0.070±0.011* | **0.004±0.001** |

Table 2: Performance comparison (**MSE @ 50 steps**) of forecasting models across SciML tasks. **Bold = best**, *italic = second and third best*.

| Task | Parrot | DynaMix | Chronos | Chronos Bolt | TimesFM | TimeMoE | Moirai |
|---|---|---|---|---|---|---|---|
| Turbulence | *0.322±0.333* | 0.490±0.4530 | 0.380±0.408 | 0.531±0.447 | 0.403±0.262 | **0.278±0.268** | **0.278±0.267** |
| ECG | *0.916±0.630* | 1.063±0.488 | 1.461±1.097 | 0.950±0.581 | 0.940±0.530 | *0.893±0.287* | **0.851±0.488** |
| Circuit | **0.012±0.016** | 0.297±0.294 | *0.024±0.030* | 0.181±0.122 | *0.065±0.056* | 0.076±0.080 | 0.075±0.060 |
| Kuramoto | **0.001±0.002** | 0.006±0.001 | 0.009±0.007 | 1.296±0.188 | 0.512±0.096 | *0.008±0.002* | **0.001±0.001** |

Table 3: Performance comparison (**KL Divergence between predicted and true attractors**) of forecasting models across SciML tasks. **Bold = best**, *italic = second and third best*.

| Task | Parrot | DynaMix | Chronos | Chronos Bolt | TimesFM | TimeMoE | Moirai |
|---|---|---|---|---|---|---|---|
| Turbulence | *0.028±0.044* | **0.005±0.008** | 0.041±0.046 | 0.048±0.058 | 0.111±0.072 | 0.070±0.058 | *0.030±0.041* |
| ECG | **0.065±0.089** | *0.099±0.104* | 0.403±0.367 | 0.253±0.185 | 0.220±0.153 | *0.188±0.094* | 0.276±0.311 |
| Circuit | *0.572±0.082* | 2.940±0.528 | *0.630±0.118* | 1.710±0.255 | **0.383±0.087** | 0.816±0.200 | 0.848±0.155 |
| Kuramoto | **0.001±0.001** | 1.010±0.150 | 0.537±0.087 | 3.116±0.202 | 4.489±0.363 | *0.076±0.040* | *0.010±0.011* |

# 6 CONCLUSION AND FUTURE DIRECTIONS

We find that a simple forecast strategy—context parroting—outperforms leading foundation models on dynamical systems forecasting, a critical task in scientific machine learning. If a foundation model cannot beat context parroting, it arguably has failed to learn the underlying physics of the system. This surprising finding exposes a limitation of current foundation models as general-purpose time-series forecasters and highlights the need to further scale them or to fine-tune them for specific domains. It also suggests that accurately measuring the performance of foundation models can be difficult for scientific machine learning tasks, because strategies like parroting can effectively game both short- and long-term accuracy metrics. Future work in SciML should measure capabilities that are orthogonal to parroting—such as inferring unobserved parameters or generalizing to unseen bifurcation regimes (Norton et al., 2025)—rather than just competing on reconstruction error.

Finding a simple but effective baseline for a challenging task can encourage rethinking of the status quo, motivating the development of better model architectures (Arora et al., 2017). For example, context parroting formalizes an explicit baseline to compare against in the time-series domain and can help discover beyond-parroting strategies. Identifying in-context learning strategies beyond parroting can spur the development of next-generation foundation models and contribute to the debate on whether (or to what extent) large language models are stochastic parrots (Bender et al., 2021; Mitchell & Krakauer, 2023; Arora & Goyal, 2023; McCoy et al., 2024).

An interesting future direction is to generalize context parroting to deal with non-stationary time series while keeping the simplicity and efficiency of the method. Context parroting assumes the existence of a stationary underlying measure; for an ergodic deterministic system this implies that conditional probabilities of timepoints are stationary up to any order (Appendix F). However, newer foundation models readily handle simple nonstationarity like baseline drift, implying that a modified parroting strategy may be possible in-context (Das et al., 2024). Once generalized, the non-stationary parroting method can replace Naive and Seasonal Naive to serve as a more informative baseline for the zero-shot forecasting of general time series (weather, traffic, finance, etc.).

Finally, we want to emphasize that we are not proposing to replace time-series foundation models with context parroting. Instead, the value of parroting is as a simple baseline that can reveal the gaps in current foundation models and guide the design of new ones. When foundation models under perform relative to context parroting, it reveals that they haven't learned to fully utilize the context data. For example, a common failure mode we observed across a range of leading foundation models (TimesFM, TimeMoE, Chronos Bolt) is that they tend to underestimate oscillations in the dynamics and the predictions often quickly converge to the mean (Appendix Fig. 6). It is also interesting to design new interpretable zero-shot forecasting strategies. For example, by combining context parroting with classical forecasting methods from the nonlinear dynamics literature (Farmer & Sidorowich, 1987; Sugihara & May, 1990).

## 7 ACKNOWLEDGMENTS

YZ acknowledges support from the Omidyar Fellowship and NSF DMS 2436231. WG was supported by NSF DMS 2436233 and NSF CMMI 2440490. This project has been made possible in part by Grant No. DAF2023-329596 from the Chan Zuckerberg Initiative DAF, an advised fund of Silicon Valley Community Foundation. Financial support for this publication results from grant CS-CSA-2026-075 from Research Corporation for Science Advancement.

## 8 REPRODUCIBILITY STATEMENT

A Python implementation of the context parroting algorithm and the benchmarks are available at https://github.com/y-z-zhang/parroting.

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

## A    SAMPLE PREDICTIONS FROM FOUNDATION MODELS

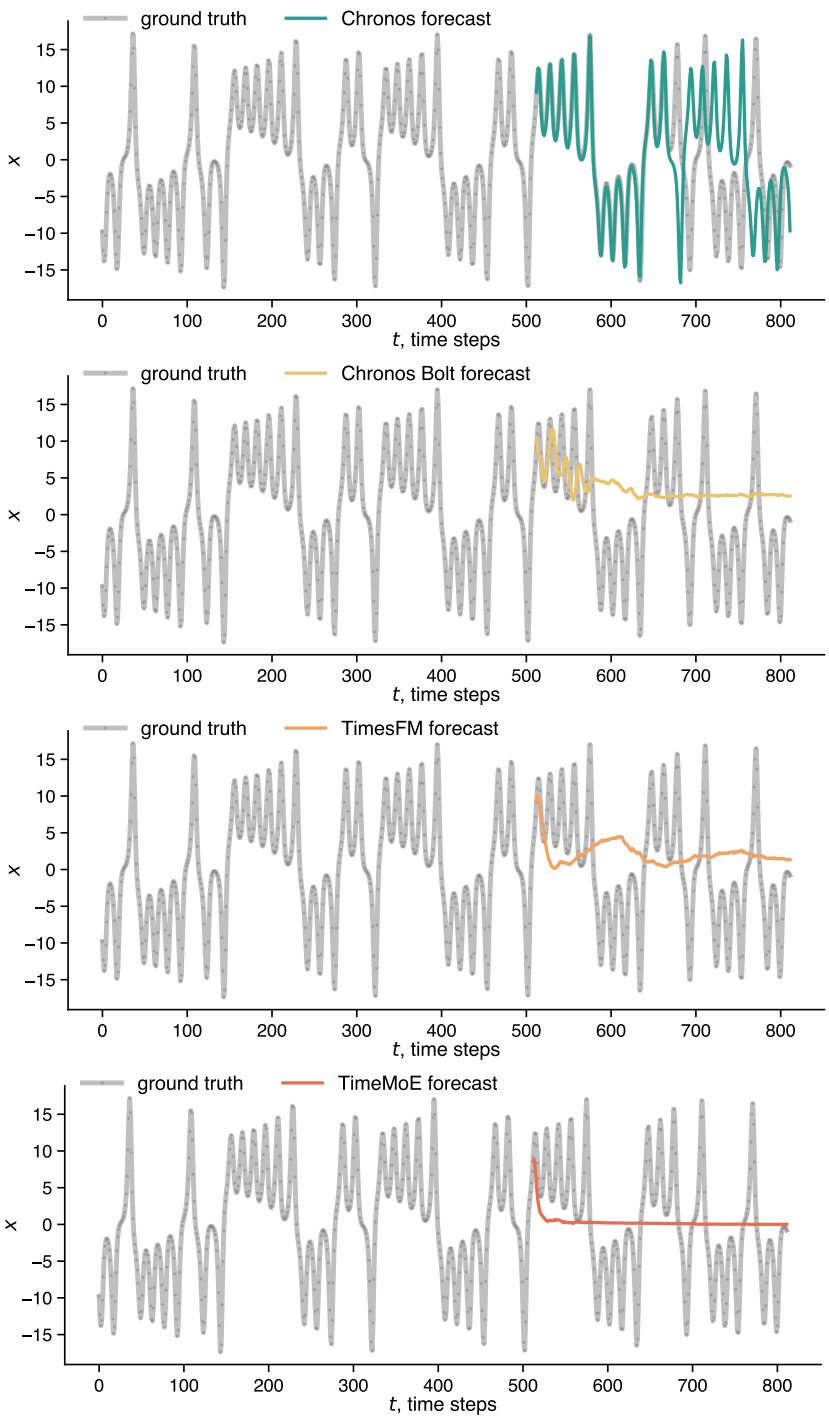

Figure 6:   **Example forecasts on a chaotic system from foundation models reveal common failure modes**. This is the same task as presented in Fig. 1 (predicting the $x$ variable of the Lorenz system based on a short context trajectory with 512 data points). Chronos does extremely well with a parroting strategy. The other models perform comparatively poorly and all exhibit a tendency to underestimate the oscillations (e.g., by quickly converging towards the mean). This is a general trend that we consistently observe across different chaotic systems and initial conditions.

## B  BENCHMARKING WITH OTHER METRICS

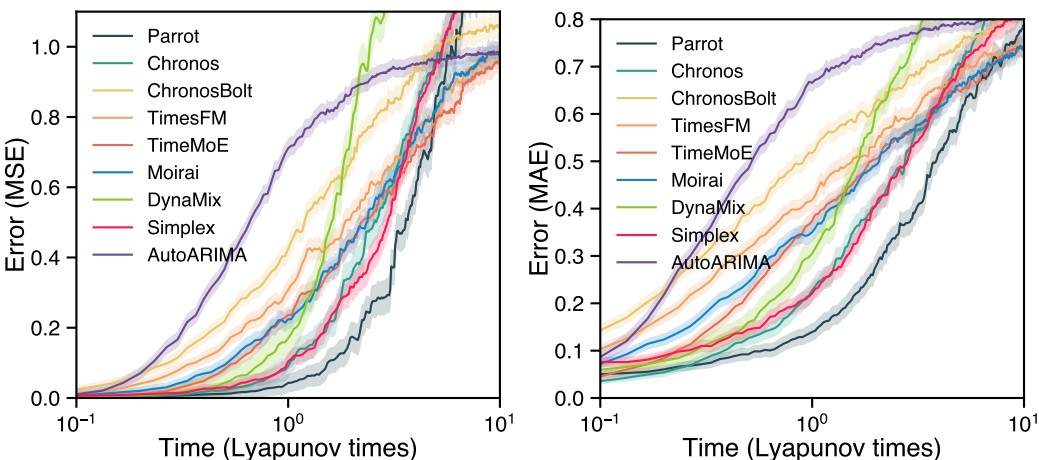

Figure 7: **Context parroting outperforms foundation models in zero-shot forecasting**. Same setup as in Fig. 2, but with the forecast error measured by MSE (left) and MAE (right). On top of the foundation models, we also include two classical forecasting methods in the comparison: simplex projection (Sugihara & May, 1990) and AutoARIMA (Hyndman & Athanasopoulos, 2018).

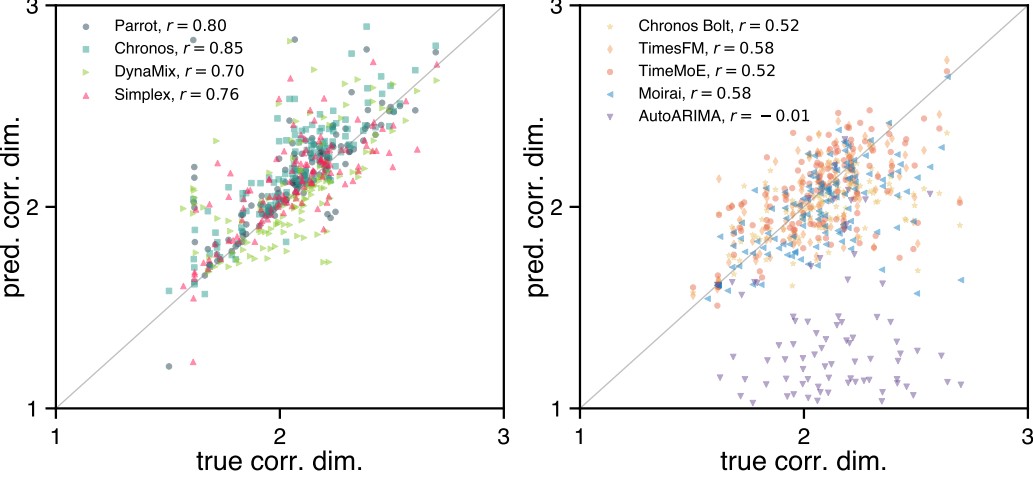

Figure 8: **Fractal dimension accuracy for parroting and foundation models**. Each point represents the predicted fractal dimension of a chaotic attractor by a model (median of 20 predictions from random initial conditions). The accuracy is measured by the Spearman correlation $r$ between the 135 predicted fractal dimensions and the true fractal dimensions.

## C  EFFECTS OF EMBEDDING DIMENSION $D$

Fig. 9 investigates how the choice of the embedding dimension $D$ affects the performance of context parroting. Overall, the valid prediction time stays consistent over a wide range of embedding dimension $D$. For short context windows, there is a slight advantage to small $D$. For long context windows, larger embedding dimensions are marginally better. This observation suggests potential improvements in the future that choose $D$ adaptively based on factors such as context length.

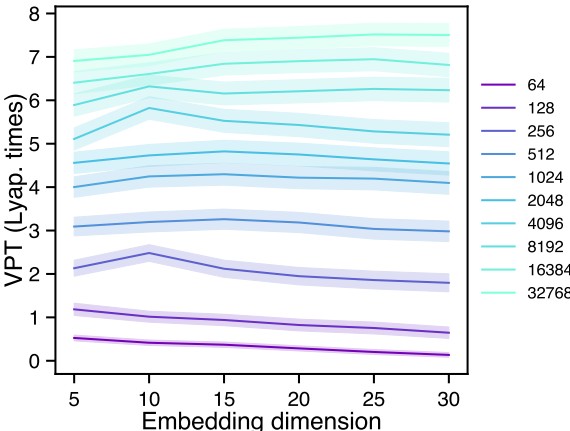

Figure 9: **Effect of the embedding dimension $D$ on the forecast accuracy of context parroting.** Each curve represents a different context length. Results are averaged over 135 chaotic systems in the `dysts` database, with 100 trajectories from random initial conditions for each system.

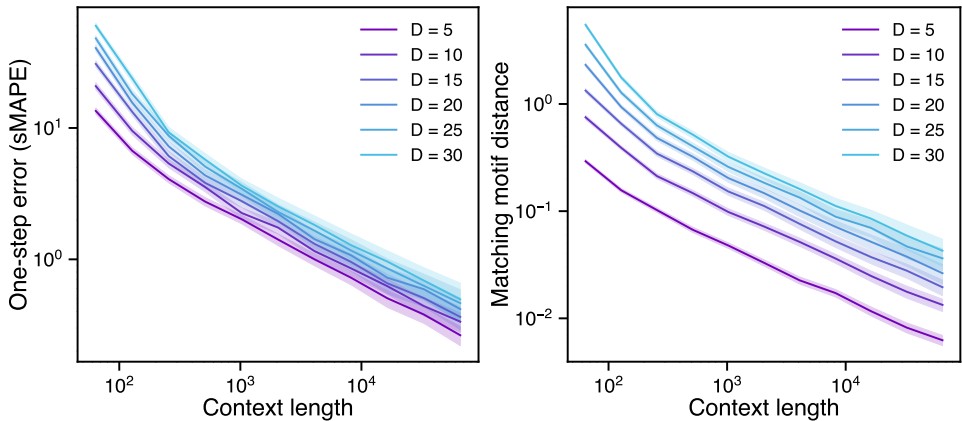

Figure 10: **Scaling laws with context length.** Left: One-step forecast error versus context length. The scaling follows a power law for all embedding dimensions $D$ considered. Smaller $D$ is more accurate here because of the one-step forecast error. Larger $D$ can be more accurate for longer forecasting horizons. Right: Euclidean distance between the last context motif $x_{L-D+1:L}$ and its closest match, as a function of context length. Again, the scaling follows a power law for all $D$.

## D   INVARIANT PROPERTIES ON LONG FORECAST HORIZONS

We next test the performance of parroting for long forecast horizons. We fix the context length $L = 512$ and then generate forecasts of length $H = 10000 - 512 = 9488$ (equivalent to over 316 Lyapunov Times). Table 4 shows the results of generating forecasts using the best-performing models from our shorter-horizon experiments. For each model, we evaluate its global accuracy by calculating (1) the correlation between the fractal dimension of the long forecast, and an estimate generated from the ground truth; (2) the correlation between the largest Lyapunov exponent of the long forecast, and an estimate generated from the ground truth; and (3) the attractor KL-divergence between the long forecast and ground truth (Grassberger & Procaccia, 1983; Rosenstein et al., 1993; Hess et al., 2023). We find that context parroting performs well on all three metrics.

| Metric | **Parrot** | **Chronos** | **Dynamix** | **Simplex** |
|---|---|---|---|---|
| Attractor KL Divergence | **0.412 ± 0.141** | 0.679 ± 0.101 | *0.508 ± 0.147* | 0.546 ± 0.140 |
| Fractal Dimension Correlation | **0.723 ± 0.042** | 0.120 ± 0.118 | *0.521 ± 0.057* | 0.341 ± 0.072 |
| Largest Lyapunov Correlation | *0.343 ± 0.018* | 0.269 ± 0.114 | **0.466 ± 0.071** | *0.343 ± 0.085* |

Table 4: KL Divergence and correlation of invariant properties between predicted and true attractors for different models for long forecast horizons. Error bars are standard deviation across all attractors for the KL Divergence, and uncertainty bounds based on the p-value for correlations. Bold = best, italic = second best.

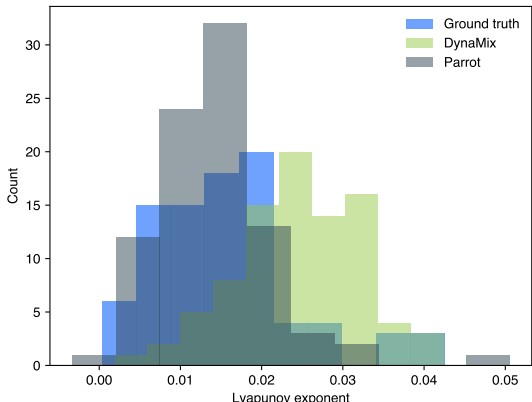

Figure 11: **Histograms of Lyapunov Exponents.** We estimate the Lyapunov exponents from the ground-truth time series, as well as from long rollouts from parroting and DynaMix, the current-generation foundation model with the best ability to preserve long-term properties. These rollouts are generated with a context length of 2000 and a prediction horizon of 10000, and correspond to estimates from all distinct chaotic systems in `dysts`.

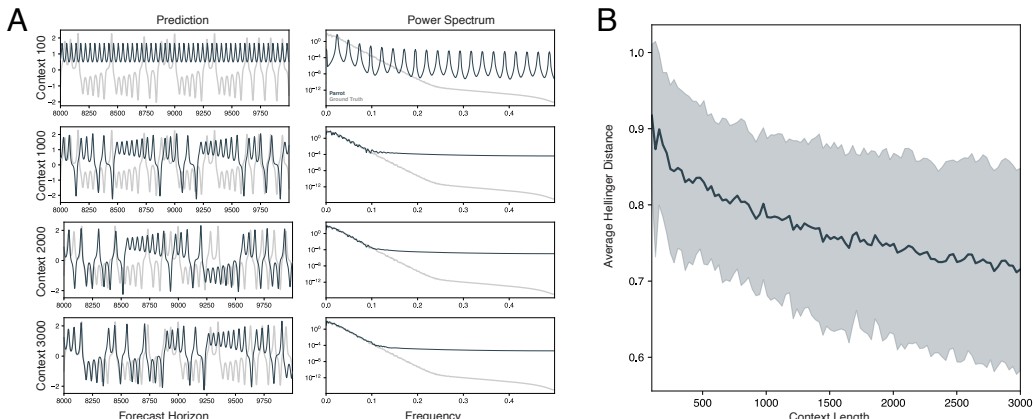

Figure 12: **Invariant properties predicted by parroting improve with context length.** (A) Predictions of a single chaotic system, the Lorenz attractor, by the parroting model as the context length is increased from 100 to 3000. Forecasts are generated for $10,000$ points past the end of the context, and the last 2000 timepoints are shown. The power spectrum is estimated using Welch's method on the last 5000 timepoints. (B) The average Hellinger distance between the true and predicted power spectrum as a function of context length, averaged over 129 distinct chaotic systems (including the Lorenz attractor). Error bars correspond to standard deviations. The averaged Hellinger distance is introduced as a long-term metric for chaotic systems in Mikhaeil et al. (2022) and Brenner et al. (2022).

## E EFFECTS OF NOISE AND SAMPLING RATE

We add Gaussian noise to normalized chaotic trajectories and repeat the original experiments (a noise level of 0.1 translates to $10\%$ perturbation on each data point on average). The results are consistent across different orders of magnitude in noise, and parroting is consistently the best or the second best in all experiments.

| Noise level | Parrot | Chronos | Chronos Bolt | TimesFM | TimeMoE |
|---|---|---|---|---|---|
| $10^{-3}$ | **2.17±0.19** | *1.68±0.18* | 0.79±0.18 | 1.07±0.19 | 0.92±0.15 |
| $10^{-2}$ | **2.10±0.18** | *1.65±0.18* | 0.79±0.18 | 1.05±0.19 | 0.92±0.16 |
| $10^{-1}$ | **1.04±0.14** | *0.89±0.15* | 0.71±0.17 | *0.89±0.17* | 0.66±0.11 |

Table 5: **Valid prediction time** across noise levels (higher is better). Bold = best, italic = second best. Shading highlights best (dark) and second best (light).

| Noise level | Parrot | Chronos | Chronos Bolt | TimesFM | TimeMoE |
|---|---|---|---|---|---|
| $10^{-3}$ | **0.233±0.221** | *0.297±0.243* | 0.491±0.223 | 0.440±0.228 | 0.377±0.211 |
| $10^{-2}$ | **0.235±0.220** | *0.311±0.245* | 0.492±0.223 | 0.441±0.228 | 0.383±0.211 |
| $10^{-1}$ | **0.286±0.209** | *0.366±0.235* | 0.509±0.219 | 0.455±0.224 | 0.415±0.216 |

Table 6: **MAE @ 1 Lyapunov Time** across noise levels (lower is better). Bold = best, italic = second best. Shading highlights best (dark) and second best (light).

| Noise level | Parrot | Chronos | Chronos Bolt | TimesFM | TimeMoE |
|---|---|---|---|---|---|
| $10^{-3}$ | **0.183±0.339** | *0.268±0.367* | 0.473±0.314 | 0.394±0.307 | 0.315±0.273 |
| $10^{-2}$ | **0.185±0.340** | *0.282±0.361* | 0.474±0.314 | 0.394±0.306 | 0.318±0.274 |
| $10^{-1}$ | **0.220±0.346** | *0.328±0.373* | 0.489±0.314 | 0.407±0.302 | 0.349±0.285 |

Table 7: **MSE @ 1 Lyapunov Time** across noise levels (lower is better). Bold = best, italic = second best. Shading highlights best (dark) and second best (light).

| Noise level | Parrot | Chronos | Chronos Bolt | TimesFM | TimeMoE |
|---|---|---|---|---|---|
| $10^{-3}$ | *0.73* | **0.85** | 0.52 | 0.58 | 0.52 |
| $10^{-2}$ | *0.63* | **0.77** | 0.51 | 0.55 | 0.45 |
| $10^{-1}$ | **0.59** | *0.57* | 0.37 | 0.49 | 0.16 |

Table 8: **Fractal dimension accuracy** (Spearman correlation) across noise levels (higher is better). Bold = best, italic = second best. Shading highlights best (dark) and second best (light).

| Noise level | Parrot | Chronos | Chronos Bolt | TimesFM | TimeMoE |
|---|---|---|---|---|---|
| $10^{-3}$ | **0.113±0.205** | *0.173±0.209* | 0.346±0.297 | 0.345±0.298 | 0.354±0.290 |
| $10^{-2}$ | **0.115±0.201** | *0.189±0.244* | 0.344±0.292 | 0.356±0.298 | 0.344±0.285 |
| $10^{-1}$ | **0.141±0.207** | *0.218±0.263* | 0.382±0.314 | 0.389±0.306 | 0.433±0.306 |

Table 9: **KL Divergence** between predicted and true attractors across noise levels (lower is better). Bold = best, italic = second best. Shading highlights best (dark) and second best (light).

In the main text, we set an intermediate granularity of 30 points per Lyapunov time. Below we compare it with results obtained for granularities of 10 points per Lyapunov time and 50 points per Lyapunov time. Granularity does not strongly affect the results or relative model ranking. Parroting is either the best or the second best in all experiments. This makes sense, as we would expect changing

granularity to have a similar effect as rescaling of time (although with bigger or smaller gaps between data points). For example, if we use finer granularity by a factor of 2, then we would need to double the context length to get the same lookback window.

| Granularity | Parrot | Chronos | Chronos Bolt | TimesFM | TimeMoE |
|---|---|---|---|---|---|
| 10 | **4.70±0.57** | *3.93±0.59* | 1.55±0.50 | 1.92±0.54 | 1.43±0.26 |
| 30 | **2.15±0.19** | *1.68±0.18* | 0.79±0.18 | 1.07±0.19 | 0.92±0.15 |
| 50 | **1.41±0.11** | *1.12±0.11* | 0.55±0.10 | 0.79±0.11 | 0.54±0.05 |

Table 10: **Valid prediction time** across different granularities (higher is better). Bold = best, italic = second best.

| Granularity | Parrot | Chronos | Chronos Bolt | TimesFM | TimeMoE |
|---|---|---|---|---|---|
| 10 | **0.219±0.204** | *0.316±0.256* | 0.567±0.226 | 0.481±0.218 | 0.414±0.232 |
| 30 | **0.233±0.221** | *0.297±0.243* | 0.491±0.223 | 0.440±0.228 | 0.377±0.211 |
| 50 | **0.270±0.226** | *0.329±0.241* | 0.527±0.224 | 0.448±0.235 | 0.412±0.193 |

Table 11: **MAE @ 1 Lyapunov Time** across different granularities (lower is better). Bold = best, italic = second best.

| Granularity | Parrot | Chronos | Chronos Bolt | TimesFM | TimeMoE |
|---|---|---|---|---|---|
| 10 | **0.163±0.311** | *0.291±0.364* | 0.565±0.314 | 0.429±0.293 | 0.349±0.300 |
| 30 | **0.164±0.295** | *0.268±0.367* | 0.473±0.314 | 0.394±0.307 | 0.315±0.273 |
| 50 | **0.224±0.347** | *0.310±0.377* | 0.536±0.341 | 0.426±0.331 | 0.352±0.272 |

Table 12: **MSE @ 1 Lyapunov Time** across different granularities (lower is better). Bold = best, italic = second best.

| Granularity | Parrot | Chronos | Chronos Bolt | TimesFM | TimeMoE |
|---|---|---|---|---|---|
| 10 | **0.87** | *0.82* | 0.34 | 0.39 | 0.36 |
| 30 | *0.80* | **0.85** | 0.52 | 0.58 | 0.52 |
| 50 | **0.89** | *0.86* | 0.41 | 0.60 | 0.56 |

Table 13: **Fractal dimension accuracy** (Spearman correlation) across different granularities (higher is better). Bold = best, italic = second best.

| Granularity | Parrot | Chronos | Chronos Bolt | TimesFM | TimeMoE |
|---|---|---|---|---|---|
| 10 | **0.087±0.137** | *0.127±0.173* | 0.573±0.307 | 0.444±0.326 | 0.467±0.368 |
| 30 | **0.122±0.194** | *0.173±0.209* | 0.346±0.297 | 0.345±0.298 | 0.354±0.290 |
| 50 | **0.137±0.207** | *0.230±0.256* | 0.406±0.305 | 0.361±0.323 | 0.370±0.338 |

Table 14: **KL Divergence** between predicted and true attractors across different granularities (lower is better). Bold = best, italic = second best.

## F  THEORETICAL PROPERTIES OF CONTEXT PARROTING

### F.1  OVERVIEW

**Mathematical Formulation.** Context parroting corresponds to a continuous 1-nearest-neighbor search over sequences of length $D$ in the context of length $L$. It thus corresponds to a limit of a

Nadaraya–Watson model of the time series,

$$\hat{p}(\mathbf{y} \mid \mathbf{q}) = \frac{\sum_{j=D}^{L-H} K_\sigma\left(\mathbf{q}, \mathbf{x}_{j-(D-1):j}\right) K_\sigma\left(\mathbf{y}, \mathbf{x}_{j+1:j+H}\right)}{\sum_{j=D}^{L-H} K_\sigma\left(\mathbf{q}, \mathbf{x}_{j-(D-1):j}\right)}, \tag{1}$$

where the query $\mathbf{q}$ represents the length-$D$ motif immediately preceding the end of the context window. $\mathbf{y}$ represents a length-$H$ forecast of subsequent values. The forecast sequence $\mathbf{y}$ has probability $\hat{p}$ conditioned on the query. The symmetric kernel $K_\sigma(\mathbf{u}, \mathbf{v}) = \sigma^{-d} K\left((\mathbf{u} - \mathbf{v})/h\sigma\right)$ has bandwidth $\sigma$ in dimension $d = D \cdot \dim(x_t)$. Assuming mean-squared error as a distance function in sequence space, we use a Gaussian kernel

$$K_\sigma(\mathbf{u}, \mathbf{v}) = \frac{1}{(2\pi\sigma^2)^{d/2}} \exp\left(-\frac{\|\mathbf{u} - \mathbf{v}\|^2}{2\sigma^2}\right)$$

We set the second kernel on $\mathbf{y}$ in Eq. 1 to a delta function, in order to output predictions that exactly match sequences from the context, rather than nearby sequences in a least-squares sense. We write the conditional mean predictor

$$\hat{\mathbf{y}}(\mathbf{q}) = \sum_{j=D}^{L-H} w(\mathbf{q}, \mathbf{x}_{j-(D-1):j})\, \mathbf{x}_{j+1:j+H}, \quad w(\mathbf{q}, \mathbf{z}) \equiv \frac{K_\sigma(\mathbf{q}, \mathbf{z})}{\sum_{j=D}^{L-H} K_\sigma\left(\mathbf{q}, \mathbf{x}_{j-(D-1):j}\right)}. \tag{2}$$

Context parroting corresponds to the 1-nearest-neighbor limit $\sigma \to 0$.

**Context parroting preserves attractor properties at long context lengths.** In Appendix F.4, we derive the following proposition,

$$\lim_{L \to \infty} \mathbb{E}_p[F(\mathbf{y})|\mathbf{q}] = \mathbb{E}_\mu[F(\mathbf{x})]$$

where $L$ is the context length for an Nadaraya–Watson estimator $p$, $F(\mathbf{y})$ is an estimate from a forecast sequence $\mathbf{y}$ of a property $F$ of an ergodic dynamical system, which has an invariant value $\mathbb{E}_\mu[F(\mathbf{x})]$ when calculated over the full attractor with underlying measure $\mu$. The query $\mathbf{q}$ is an arbitrary sequence of consecutive timepoints from the dynamical system. This proposition states that, when the context is sufficiently long, context parroting of an ergodic system preserves invariant values of the underlying dynamics. Context parroting thus represents an effective baseline for dynamical systems forecasting, because, in the limit of long context, it will preserve global properties like conditional distributions of values, Lyapunov exponents, or entropy production rates.

## F.2 DISCRETE-TOKEN PARROTING

For fully-discrete tokens, a $D^{th}$ order Markov chain fit to the context has the form

$$p(\mathbf{y}|\mathbf{q}) = \frac{\#\{j : (x_{j-(D-1):j} = \mathbf{q}) \wedge (x_{j+1:j+H} = \mathbf{y})\}}{\sum_{y'} \#\{i : (x_{j-(D-1):j} = \mathbf{q}) \wedge (x_{j+1:j+H} = \mathbf{y}')\}} \tag{3}$$

where the overall context has length $L$, and the Markov chain conditions the $H < L$ future tokens on the $D < L$ preceding tokens. The index $j$ runs over all contiguous sequences of length $D + H$ in the context, $j \in \{D - 1, D, ..., L - H - 2, L - H - 1\}$. The vector $\mathbf{q} \in \mathbb{R}^D$ represents the query, and the vector $\mathbf{y} \in \mathbb{R}^H$ represents the prediction in response to this query. Eq. 3 simply counts the number of token sequences of length $D + H$ that start with a given sequence of $D$ query tokens. A maximum-likelihood estimator derived from this model always samples the highest-likelihood sequence $\mathbf{y}$,

$$\hat{\mathbf{y}}_{\text{MLE}}(\mathbf{q}) = \text{argmax}_{\mathbf{y}} \log p(\mathbf{y}|\mathbf{q})$$

However, this estimator may be unstable due to the appearance of queries $\mathbf{q}$ not seen in the context, motivating the use of *token smoothing*, in which Eq. 3 is replaced by the distribution

$$p(\mathbf{y}|\mathbf{q}) = \frac{\#\{j : (x_{j-(D-1):j} = \mathbf{q}) \wedge (x_{j+1:j+H} = \mathbf{y}) + \alpha}{\sum_{\mathbf{y}'} \left(\#\{i : (x_{j-(D-1):j} = \mathbf{q}) \wedge (x_{j+1:j+H} = \mathbf{y}') + \alpha\right)} \tag{4}$$

with increasing values of the parameter $\alpha$ causing predictions to converge to a uniform sample over possible predictions $\mathbf{y}$. The parameter value $\alpha = 0$ reduces to no smoothing, while $\alpha = 0.5$ corresponds to the Jeffreys prior and $\alpha = 1$ corresponds to Laplace's rule of succession.

### F.3 CONTINUOUS-TOKEN PARROTING

A more general time series model treats tokens as continuous-valued. Some time series foundation models like Chronos use binning to discretize time series values, allowing the direct use of discrete-token architectures (Ansari et al., 2024). However, many time series models assume effective continuity in token values, and we favor a continuous formulation in order to highlight connections to dynamical systems theory.

To model continuous-valued tokens directly, we replace the discrete count in §F.2 with a kernel-weighted estimate over all past subsequences. Let $\{\mathbf{x}_t\}$ denote a univariate or multivariate time series. For context length $L$ and prediction horizon $H$, the Nadaraya–Watson estimate of the conditional density is

$$\hat{p}(\mathbf{y} \mid \mathbf{q}) = \frac{\sum_{j=D}^{L-H} K_h\left(\mathbf{q}, \mathbf{x}_{j-(D-1):j}\right) \; K_h\left(\mathbf{y}, \mathbf{x}_{j+1:j+H}\right)}{\sum_{j=D}^{L-H} K_h\left(\mathbf{q}, \mathbf{x}_{j-(D-1):j}\right)}, \tag{5}$$

where $K_h(\mathbf{u}, \mathbf{v}) = h^{-d} K\left((\mathbf{u} - \mathbf{v})/h\right)$ is a symmetric kernel with bandwidth $h$ in dimension $d = D \cdot \dim(\mathbf{x}_t)$ for the first kernel, and $d = H \cdot \dim(\mathbf{x}_t)$ for the second kernel. Assuming mean-squared error as a distance function in sequence space, we use a Gaussian kernel

$$K_h(\mathbf{u}, \mathbf{v}) = \frac{1}{(2\pi h^2)^{d/2}} \exp\left(-\frac{\|\mathbf{u} - \mathbf{v}\|^2}{2h^2}\right)$$

In practice, we drop the second kernel on $\mathbf{y}$ in Eq. 5 in order to output a prediction that exactly matches sequences from the context, rather than nearby sequences in a least-squares sense. We thus write the conditional mean predictor

$$\hat{\mathbf{y}}(\mathbf{q}) = \sum_{j=D}^{L-H} w(\mathbf{q}, \mathbf{x}_{j-(D-1):j}) \, \mathbf{x}_{j+1:j+H} \tag{6}$$

where we have isolated a term corresponding to the weight of each sequence,

$$w(\mathbf{q}, \mathbf{z}) \equiv \frac{K_h\left(\mathbf{q}, \mathbf{z}\right)}{\sum_{j=D}^{L-H} K_h\left(\mathbf{q}, \mathbf{x}_{j-(D-1):j}\right)}.$$

**Nearest-neighbor and global average limits.** The bandwidth $h$ plays the role of a smoothing parameter (analogous to $\alpha$ in Eq. 4). As $h \to 0$ the scheme approximates a single-nearest neighbor parrot, while as $h \to \infty$ it converges to a global average over all sequences.

**Connection to attention.** If one takes

$$K(\mathbf{u}, \mathbf{v}) = \exp\left(\mathbf{u}^\top \mathbf{v}/\tau\right),$$

then Eq. 6 recovers a simplified form of softmax-attention, with the temperature hyperparameter $\tau$ controlling smoothness. In this view, the continuous parroting scheme is a kernel-regression analogue of discrete $k$–gram smoothing (Tsai et al., 2019).

**k-nearest-neighbor limit.** We define a set Top$k$ corresponding to a subset of the possible values of the index $j \in \{D, D+1, ..., L-H-1, L-H\}$. The $k$ elements of Top$k$ correspond to the indices $j$ that produce the $k$ largest values of $w(\mathbf{q}, \mathbf{x}_{j-(D-1):j})$ across all values of $j$. We compute a simple average of these $k$ closest matches

$$\hat{\mathbf{y}}(\mathbf{q}) = \frac{1}{k} \sum_{j \in \text{Top}k} w(\mathbf{q}, \mathbf{x}_{j-(D-1):j}) \, \mathbf{x}_{j+1:j+H} \tag{7}$$

yielding a $k$–nearest-neighbors parroting scheme. As $k$ increases, this estimator interpolates between exact parroting ($k = 1$) and global average ($k \to L$).

**Simplex projection.** Simplex projection, a classical forecasting method in nonlinear dynamics, corresponds to the special case $H = 1$ (single step prediction), $k = D + 1$ in Eq. 7. The condition $k = D + 1$ represents the minimal number of affinely independent neighbors needed to triangulate a point in a $D$-dimensional space (Sugihara & May, 1990).

In simplex projection, the query $\mathbf{q}$ is interpreted as a time-delay embedding of the time series observable $\mathbf{x}$. Takens' theorem argues that, under mild conditions, a finite number of time delay embeddings of an observable drawn from a deterministic ergodic system will be diffeomorphic (smoothly mappable) to the full-state dynamics (Takens, 2006). Because simplex projection uses only neighbor identities, and not absolute distances, to weight context points, a delay embedding is sufficient to calculate the appropriate weights.

**S-map forecasts.** Another common nonlinear forecasting technique retains all terms in the sum Eq. 6 , but instead performs a nonlinear weighting of the form

$$K_\theta(\mathbf{u}, \mathbf{v}) = \exp(-\theta\|\mathbf{u} - \mathbf{v}\|/\bar{d})$$

where the scale parameter $\bar{d}$ is determined by the distribution of distances among queries and points in the context. In practice, this parameter is often set to the mean pairwise distance among all sequences of length $D$ in the context. The optimal value of the hyperparameter $\theta$ increases as the underlying dynamics become more strongly nonlinear Sugihara (1994). We note that, in the classical formulation of the S-map, a locally-linear model is fit based on all sequences of length $D + H$ seen in the context, while here we use the Nadaraya–Watson estimator in order to emphasize connections to modern kernel regression.

## F.4 INVARIANTS OF MOTION

For ergodic dynamical systems in continuous time, there exists a natural measure $\mu(\mathbf{x})$ such that, for certain observables $F(\mathbf{x})$, the following condition almost surely holds,

$$\mathbb{E}_\mu[F] \equiv \lim_{T\to\infty} \frac{1}{T} \int_0^T F(\mathbf{x}) = \int F(\mathbf{x})d\mu(\mathbf{x}) = \text{constant}$$

where the second equality arises from the Birkhoff ergodic theorem (Walters, 1982).

We use the following convention for expectation values of sequences and single tokens; the expectation $\mathbb{E}_\mu[\mathbf{x}_{t:t+T}]$ refers to the expected value of the sequence $\mathbf{x}_{t:t+T}$ given pointwise measure $\mu$. We note that, for deterministic dynamical systems, once a given point is sampled on the attractor with measure $\mu(\mathbf{x}_t)$, subsequent points have delta function conditional probability on the first point. Thus, we use the convention $\mu(\mathbf{x}_t) = \mu(\mathbf{x}_{t:t+T})$ and we use the measure to refer to both the probability of a given timepoint, or a sequence of arbitrary length originating from that timepoint.

**Proposition.** Under appropriate kernel conditions,

$$\lim_{L\to\infty} \mathbb{E}_p[F(\mathbf{y})|\mathbf{q}] = \mathbb{E}_\mu[F(\mathbf{x})]$$

where $L$ is the context length for a Nadaraya–Watson estimator $p$, $F(\mathbf{y})$ is an estimate on a sequence $\mathbf{y}$ of an invariant property of an ergodic dynamical system with measure $\mu$, and $\mathbf{q}$ is an arbitrary sequence of consecutive timepoints from the dynamical system. This proposition states that, when the context is sufficiently long, a Nadaraya–Watson estimator of an ergodic system preserves the invariant values of the underlying dynamics.

**Derivation.** We start with the definition of the dynamical average,

$$\mathbb{E}_\mu[F] = \int F(\mathbf{x})d\mu(\mathbf{x})$$

Inserting Eq. 5 into this expression,

$$\mathbb{E}_\mu[F(\mathbf{y})|\mathbf{q}] = \frac{\sum_{j=D}^{L-H} K_h\left(\mathbf{q}, \mathbf{x}_{j-(D-1):j}\right) \int F(\mathbf{y}) K_h\left(\mathbf{y}, \mathbf{x}_{j+1:j+H}\right) d\mu(\mathbf{y})}{\sum_{j=D}^{L-H} K_h\left(\mathbf{q}, \mathbf{x}_{j-(D-1):j}\right)},$$

We multiply both the numerator and denominator by $1/L$ and take the limit $L \to \infty$, in order to convert the summations to expectations,

$$\lim_{L\to\infty} \mathbb{E}_\mu[F(\mathbf{y})|\mathbf{q}] = \frac{\mathbb{E}_\mu\left[K_h(\mathbf{q}, \mathbf{x}_\leftarrow) \int F(\mathbf{y}) K_h\left(\mathbf{y}, \mathbf{x}_\rightarrow\right) d\mu(\mathbf{y})\right]}{\mathbb{E}_\mu[K_h(\mathbf{q}, \mathbf{x}_\leftarrow)]},$$

where $\mathbf{x}_{\leftarrow}$ represents the first $D$ points of random lookback window of length $D + H$ sampled from the underlying dynamical system, while $\mathbf{x}_{\rightarrow}$ denotes the next $H$ timepoints. In practice, this corresponds to a time series of $D + H$ points generated by simulating the dynamics starting at a point on the attractor randomly-sampled according to the measure $\mu$.

If we take the limit $h \to 0$ (exact matching), then the kernel $K_h$ becomes a delta function, yielding

$$\lim_{h \to 0} \lim_{L \to \infty} \mathbb{E}_\mu[F(\mathbf{y})|\mathbf{q}] = \mathbb{E}_\mu[F(\mathbf{x}_{\rightarrow})|\mathbf{x}_{\leftarrow} = \mathbf{q}]$$

If the measure $\mu$ is ergodic, then the conditional expectation of an invariant $F$ given any query $\mathbf{q}$ is simply its unconditional expectation,

$$\lim_{h \to 0} \lim_{L \to \infty} \mathbb{E}_\mu[F(\mathbf{y})|\mathbf{q}] = \mathbb{E}_\mu[F(\mathbf{x})]$$

## F.5 SCALING LAWS LIMITING PREDICTION OF STOCHASTIC SYSTEMS

For a stochastic time series $\mathbf{x}_{1:T}$ with autocorrelation given by

$$|\text{Corr}(\mathbf{x}_t, \mathbf{x}_{t+\tau})| \leq Ce^{-\alpha\tau}, \quad \alpha > 0$$

with $C$ representing a proportionality constant, the expected mean squared error of a forecast scales as

$$\mathbb{E}\big[\|\hat{\mathbf{y}} - \mathbf{y}\|^2\big] \sim e^{-\alpha L}, \quad L \to \infty.$$

Thus, under exponential decay of correlations (mixing), the amount of information about future states in a length-$L$ context window saturates exponentially quickly Bradley (2005). Thus, forecasts derived from increasingly large context windows converge exponentially quickly to optimal conditional forecasts under the invariant measure $\mu$.

Under standard smoothness conditions Fan & Yao (2008); Takezawa (2005), the forecast error also exhibits a standard bias-variance tradeoff of the form

$$\mathbb{E}\big[\|\hat{\mathbf{y}} - \mathbf{y}\|^2\big] = \mathcal{O}(h^4) + \mathcal{O}\left(\frac{1}{Ch^{L+H}}\right).$$

The optimal width of the kernel thus scales as,

$$h_{\text{opt}} \sim C^{-1/(4+L+H)}$$

