# OpenReview forum: "Context parroting: A simple but tough-to-beat baseline for foundation models in scientific machine learning"
_ICLR.cc/2026/Conference — ICLR 2026 Poster_

### Official Review · Reviewer_rUtK · 2025-10-14

**Soundness:** 4
**Presentation:** 4
**Contribution:** 2
**Rating:** 8
**Confidence:** 4

**Summary:**

# Review for Context Parroting

## Summary
- This paper presents parroting as a baseline for foundation models in scientific ML. The authors demonstrate that a simple algorithm outperforms foundation models in zero-shot forecasting of chaotic dynamical systems. They perform several analysis, comparing short- and long-time forecasts and motivate the power scaling law in LLMs with respect to one-step error and context length. While the technical contribution is simple, it does demonstrate as a useful baseline for comparing foundation models on forecasting chaotic systems.

## Recommendation
- Accept given the below feedback and questions are addressed. Particularly, provide a clean repository of code for reproducability (should not be too much effort) and add more comparisons (more models like Moirai and/or LLMTime as well as the classical methods described in the paper, and maybe even comparing sampling strategies).

**Strengths:**

## Strengths
- The necessary dynamics backgrounds are explained well and sufficient detail (eg. Lyapunov time)
- The parroting method is clearly explained.
- The computational experiment for comparing models is explained clearly.
- The evaluation metrics sMAPE and KL divergence are clearly explained.
- The results are compelling and clearly laid out.

**Weaknesses:**

## Weaknesses
- The reproducability is lacking. There should be a GitHub page with code that is easily run to recreate the figures and experiments.
- The authors mention classical forecasting methods like Simplex projection and S-map forecasts and connect it to context parroting, however, these classical methods are not used in any comparisons.

**Questions:**

## Questions
- Can you clarify the difference between fractal dimension, correlation dimension, and scaling coefficient? In the paper it's not clear why fractal dimension and correlation dimension are used separately when they appear to mean the same thing, this makes it slightly confusing.
- I'm curious how different sampling strategies affect LLM results (eg. beam search, top-p, top-k, etc..). Please clarify if sampling strategy will affect results and how so.
- I would like to see how off-the-shelf LLMs like Llama perform on this task too (see Gruver (2023)) [[link]](https://proceedings.neurips.cc/paper_files/paper/2023/file/3eb7ca52e8207697361b2c0fb3926511-Paper-Conference.pdf). Was this considered?
- Classical methods are mentioned and a connection is drawn to parroting, why not just implement them and compare the results? How do those methods compare with parroting?
- Why not also test multivariate foundation models like Moirai?

## Feedback
- Elaborate on the parenthetical 'can we estimate the "fractal dimension" of a language' and the potential applications there (line 099). Consider moving this to the conclusion (section 6).
- Provide a direct citation of Takens':
```
@inproceedings{takens2006detecting,
  title={Detecting strange attractors in turbulence},
  author={Takens, Floris},
  booktitle={Dynamical Systems and Turbulence, Warwick 1980: proceedings of a symposium held at the University of Warwick 1979/80},
  pages={366--381},
  year={2006},
  organization={Springer}
}
```

- Cite the Chronos Time-MoE and TimesFM models in "Models." (line 196)
- Axes of figure 2 right need to be explained more. What do the widths mean? (line 275)
- Elaborate more on why Chronos has better correlation dimension that parroting.
- Line 383, "This is equivalent to embed" to "this is equivalent to embedding"
- Figure 4, line 371, "because the use of the" -> "because of the"
- Line 379, "Why do...?", remove this sentence. Looks unprofessional, just state the answer to the question.
- Please please please provide code to the broader community! None of this is proprietary and there should be a GitHub.

---

> ### Author Response · Authors · 2025-11-25
>
> We appreciate the reviewer's strong endorsement of our paper and thank the reviewer for their insightful comments.
>
> **[Weakness 1: Reproducibility]** We agree completely. Here is a link to the anonymized GitHub, which we will replace with a public link after the review period: <https://anonymous.4open.science/r/parroting-4D26>. We have updated the "Reproducibility Statement" accordingly.
>
> **[Weakness 2: Classical methods]** Thank you for suggesting this. We have added two classical forecasting methods as new baselines: simplex projection and AutoARIMA. Context parroting also outperforms these classical methods on the dysts and SciML datasets. Please see the updated Figures 6, 7, 12, and Table 4.
>
> **[Question 1: Correlation/Fractal Dimension]** Thank you for pointing this out. We use the terms interchangeably. We have modified the text to use only "Fractal Dimension," except when we specify that the specific fractal dimension we calculate is the correlation dimension. We have also added a reference for the estimator that we use.
>
> **[Question 2: Sampling Strategies]** This is a bit tricky to benchmark, because a lot of models like the Chronos family and Moirai-2 now output quantiles (though Chronos-1 and Moirai-1 can generate samples, so probability distributions can be estimated). Our expectation is that context parroting is a high-variance estimator, it essentially picks a single match from the data and commits to repeating it, making it inherit the properties of a top-1 estimator compared to top-k (higher bias, lower variance estimators). As a result, models with parroting tendencies like Chronos-1 will make some very accurate forecasts, but also some very inaccurate ones, resulting in decent but highly-varying scores across different forecasts. In contrast, models that perform some kind of nucleus search would average over different context-matches, resulting in mean regression (Chronos Bolt tends to do this). This looks okay on many metrics, but the forecasts rarely look strongly right.
>
> **[Question 3: off-the-shelf LLMs]** We have added benchmarks on more recent foundation models: Dynamix (designed specifically for dynamical systems) and Moirai-2. See, for example, Figure 2 and Tables 1-3. It was shown previously [1] that foundation models trained on time series data in general outperform off-the-shelf LLMs on forecasting tasks. For this reason we did not include off-the-shelf LLMs in our comparison. A previous study did apply off-the-shelf LLMs to dynamical systems forecasting tasks and found interesting in-context neural scaling laws [2]. Our paper extends this work and offers an explanation of the in-context neural scaling law based on context parroting.
>
> [1]: Ansari, et al. "Chronos: Learning the language of time series." arXiv:2403.07815 (2024).
>
> [2]: Liu, et al. "LLMs learn governing principles of dynamical systems, revealing an in-context neural scaling law." arXiv:2402.00795 (2024).
>
> **[Question 4: More baselines]** We have added two classical forecasting methods as new baselines: simplex projection and AutoARIMA, as well as multivariate foundation models like Moirai-2. See our responses above.
>
> **[Feedback]** Thank you, we have implemented most of the suggestions. We also released the code (currently as an anonymized GitHub repo link).
>
> We thank the reviewer again for their insightful comments. Please let us know if there are additional questions that we can help address.

---

### Official Review · Reviewer_2L3R · 2025-10-25

**Soundness:** 4
**Presentation:** 3
**Contribution:** 3
**Rating:** 10
**Confidence:** 5

**Summary:**

This paper investigates context parroting, a simple “copying” strategy that mimics motifs from past context, as a surprisingly strong baseline for zero-shot forecasting in scientific machine learning (SciML). The authors show that this trivial method can outperform several state-of-the-art time-series foundation models (Chronos, Chronos-Bolt, TimesFM, Time-MoE) on forecasting chaotic and physical systems. They connect this behavior to induction heads in transformers and to in-context neural scaling laws, deriving a geometric explanation that ties scaling exponents to the fractal dimension of chaotic attractors. The work suggests that many current foundation models rely heavily on parroting-like mechanisms, calling for benchmarks and architectures that go beyond this trivial solution.

**Strengths:**

Strong empirical evidence: Extensive benchmarking across 135 chaotic systems and multiple real-world datasets (e.g., ECG, Kuramoto oscillators) convincingly supports the claim that context parroting is competitive or superior to foundation models.

Clarity and interpretability: The algorithm is simple, well-motivated, and connected to classic nonlinear dynamics methods (Takens’ embedding, simplex projection).

Insightful theoretical link: The explanation of in-context scaling laws through the fractal dimension of attractors is elegant and potentially generalizable beyond SciML.

Critical perspective: The paper exposes fundamental limitations of current time-series foundation models—especially their tendency to “regress to the mean” and fail to utilize context effectively.

Relevance and timeliness: Given the explosion of foundation models for science, a “simple but tough-to-beat” baseline is highly valuable.

**Weaknesses:**

Scope of comparison: While the benchmarks are extensive, all compared models are pre-2025 foundations. It’s unclear whether next-generation models (e.g., fine-tuned or physics-aware transformers) would still underperform.

Limited theoretical rigor: The link between fractal dimension and scaling exponent is compelling but heuristic; a more formal derivation or empirical validation of α ≈ 1/d₍cor₎ across systems would strengthen the claim.

Potential overstatement: The claim that parroting “outperforms all foundation models” might be dataset-dependent—some tasks (e.g., short-context nonstationary regimes) still favor Chronos.

Computational fairness: The comparison omits any fine-tuning or adaptation of foundation models; parroting benefits from zero training cost but might not scale as easily to multivariate or stochastic systems.

No ablation on motif length (D) or distance metrics: Although the authors mention robustness to D, a systematic analysis would clarify when parroting breaks down.

**Questions:**

How does context parroting perform on nonstationary or regime-shifting time series (e.g., climate or economic data)?

Could a hybrid “parrot + predictor” model be constructed to retain parroting’s simplicity while addressing its limitations?

Have you tried comparing against autoregressive baselines (ARIMA, GP kernels, etc.) to quantify parroting’s advantage beyond foundation models?

Does the observed scaling α ≈ 1/d₍cor₎ persist under noise or partial observability?

Could the parroting mechanism be explicitly detected within transformer attention patterns (e.g., induction heads over repeated motifs)?

---

> ### Author Response · Authors · 2025-11-25
>
> We appreciate the reviewer's strong endorsement of our paper and thank the reviewer for their insightful comments.
>
> **[Weakness 1: Scope]** Thank you for pointing this out. We have now included more recent foundation models, such as Moirai-2 (general multivariate) and DynaMix (specifically designed for dynamical systems). All figures and tables have been updated accordingly to include these new baselines. See, for example, Figure 2 and Tables 1-3. Parroting is still among the top-performing models on almost all tasks and metrics.
>
> **[Weakness 2: Theoretical rigor]** We have edited our heuristic argument to improve its clarity. We did not include a formal derivation here because a similar scaling law has been derived in a previous study [1]. Thus, we focused on the formal description of other aspects of the parroting method, as detailed in **Appendix H: Theoretical Properties of Context Parroting**.
>
> [1] Farmer and Sidorowich. "Predicting chaotic time series." Physical Review Letters (1987).
>
> **[Weakness 3: Overstatement]** We went over the text to make sure that we avoid claims such as "parroting outperforms all foundation models". Indeed, Chronos can perform favorably in short-context nonstationary regimes, which we highlight in Page 7.
>
> **[Weakness 4: Computational fairness]** This is a good point. We focused on univariate forecast because this is a limitation for many current time-series foundation models. Parroting can scale easily to multivariate settings by matching motifs in all dimensions, and the computational cost should scale linearly with the dimensionality.
>
> **[Weakness 5: Ablation]** Please see Figure 10 for results on how the valid prediction time depends on the motif length $D$, which reveals interesting patterns. For example, for small context windows, a short motif (small $D$) is preferable; for large context windows, a long motif (large $D$) is more advantageous. This is intuitive: when context data is limited, we want to prioritize finding a match. For long context, we want to find the most informative match (i.e., motifs that coincide on many data points).
>
> **[Question 1: Nonstationarity]** We agree with the reviewer that context parroting will fail as the data becomes more nonstationary, since the procedure implicitly assumes the existence of an ergodic measure. A previous study [2] included experiments showing that Chronos-1 drops off in accuracy on a modification of the dysts dataset with induced nonstationarity. We are compute limited right now running the other requested experiments, and so we can't yet run these experiments on that dataset for all of the baselines, but we suspect this will be the case. For now, we have modified the discussion to note that stationarity is a strong assumption, and we will add these results when they are ready. We expect that all models will degrade on nonstationary data, but that newer models like TimesFM will perform comparatively better.
>
> [2] Zhang and Gilpin. "Zero-shot forecasting of chaotic systems." ICLR (2025).
>
> **[Question 2: Hybrid Model]** This is a great point! One direction we are exploring is combining parroting with a probabilistic model, such as a Gaussian Process, to account for nonstationary trends. Additionally, we also plan to improve the diversity of long-term forecasts from parroting by allowing stochastic selection among multiple matching motifs, or by using beam search / nucleus sampling. If the reviewer has a good predictor in mind for the hybrid model, we would love to hear more about it.
>
> **[Question 3: Baselines]** We agree, thank you. We have added two baseline models fully-trained on the context: AutoARIMA (using AIC for model order selection), and an implementation of Sugihara and May's "simplex projection" forecasting procedure. Please see the new Figures 6 and 7, as well as Table 4.
>
> **[Question 4: Noise and partial observability]** The systems we test are all partially observed (i.e., we forecast each channel independently), and our theoretical argument does not make assumptions on full observability. Thus, the relation $\alpha \approx 1/d_{corr}$ holds for partial observability. We haven't tested the effect of noise on scaling explicitly, but given that the reasoning is geometrical in nature, we expect the relation $\alpha \approx 1/d_{corr}$ to persist at least for weak noise.
>
> **[Question 5: Attention patterns]** We haven't yet been able to directly find signatures of induction heads in the raw attention activations. This has been harder than expected because the multihead architectures distribute patterns across many heads. We think that we'd be most likely to find this in Chronos-1 (since that uses the T5 LLM internally). We fully agree that this plus other mechanistic interpretability probes would be a good direction for future work.
>
> We thank the reviewer again for their insightful comments. Please let us know if there are additional questions that we can help address.

---

### Official Review · Reviewer_PSB6 · 2025-10-28

**Soundness:** 3
**Presentation:** 3
**Contribution:** 3
**Rating:** 6
**Confidence:** 4

**Summary:**

This paper shows that simple parroting - the process of identifying the best-matching patterns in history and repeating them as forecast - could beat state of the art time-series foundation models in predicting the evolutions of systems that are deterministic and chaotic. It then observes that simple parroting results in a scaling law between number of time steps and error in prediction quality, which is characteristic of LLMs. Finally, the paper speculates that such in-context parroting mechanism could potentially explain the in-context neural scaling of LLMs.

**Strengths:**

- The paper is well-written. It makes a convincing case for context-parroting as a simple yet non-trivial baseline for zero-shot forecast of chaotic systems.
- The paper is also very topical as it attempts to elucidate the mechanisms underlying in-context learning of foundation models, though not full achieving a mechanistic interpretation of foundation models.
- The theoretical analysis bridging the algebraic coefficient of the power law to the embedding dimension of the time series is very insightful, though only speculative.

**Weaknesses:**

Limited generality. The paper claims to explain the in-context neural scaling law reported in Liu et al. 2024 [1]. However, the in-context neural scaling law of that paper encompasses discrete, stochastic systems such as Markov chains with randomly generated transition matrices, chaotic systems such as the logistic map injected with noise, as well as deterministic ODE systems such as the Lorenz system. On the other hand, this paper (apart from section 5.3) only investigates deterministic systems governed by ordinary differential equations (ODEs). It is unclear whether context parroting could work on stochastic systems, and if so, whether it would result in similar scaling law. The paper defers the theoretical analysis of context parroting on stochastic system to future works, a valid decision in my opinion since the paper is already quite substantial and self-contained as it is. However, I still think the authors should clarify that they are only addressing the in-context neural scaling law that specifically results from learning deterministic chaotic systems governed by ODEs.

[1]: Toni JB Liu, Nicolas Boullé, Raphaël Sarfati, and Christopher J Earls. Llms learn governing principles of dynamical systems, revealing an in-context neural scaling law. arXiv:2402.00795, 2024.

**Questions:**

- The theoretical result that the power law coefficient $\alpha$ should equal $\frac{1}{d_{correlation}}$ is quite nice. However, the experimental support for this turns out to be quite weak (Fig. 11). Could the authors speculate on why?
- In Figure 5, context-parroting with more delayed embeddings correspond to curves with higher loss. This is counter-intuitive to me since more delayed embedding should result in better matching of patterns, at least when the context length is sufficiently long. However, this is not what we see in the left panel of Figure 4. Could the authors comment on why? The caption to figure 4 states "Larger D can be more accurate for longer forecasting horizons". Could the authors extend on the context length further to see whether algorithms with large D eventually outperform those with lower D?

---

> ### Author Response · Authors · 2025-11-25
>
> We thank the reviewer for their constructive feedback. We hope the reviewer will find that their comments were addressed substantively:
>
> **[Weakness 1: Limited generality]** We are glad that the reviewer agree with our decision to defer the theoretical analysis of context parroting on stochastic systems to future works. Whether the same in-context neural scaling law can be reproduced by parroting stochastic systems is definitely an interesting question worth more systematic investigation. Our intuition is that at least for the weak noise limit, parroting will still work (see Tables 5 to 9 on experiments with different levels of noise, up to 10%). Although we focused on ODE systems in our experiments, we expect parroting to also work for discrete maps such as Logistic map, at least when noise is not too large. We have modified the text accordingly to include these discussions and point out explicitly stochastic systems as a promising direction for future research. Note that the theoretical results in **Appendix H: Theoretical Properties of Context Parroting** ought to apply to the case of stochastic systems with weak noise, particularly if we relax the assumption of exact matching (where the second kernel in Eq. 1 goes to a delta function), thus allowing "fuzzy" matches in a least-squares sense.
>
> **[Q1: Scaling Estimator]** Thanks for pointing this out. The issue is the fractal dimension estimator (the Grassberger-Procaccia algorithm). Estimating the correlation dimension requires fitting a power law to an empirical curve showing the number of neighbors around each point as a function of distance. The problem is that power law fits are notoriously fragile to generate from finite-resolution time-series data [1]. As a result, the estimated scaling coefficient can vary significantly with granularity or hyperparameters of the power law estimator, and it will not necessarily converge with more data. This is an intrinsic problem with this calculation. However, in the DS community, correlation dimension is nonetheless considered one of the more reliable data-driven invariant quantities that can be estimated directly from time series (data-driven Lyapunov exponent estimators such as the Rosenstein algorithm are even more fragile). We've revised the paper to clarify that the numerical challenge is a limitation of the metric, not the dataset or experiments themselves. We have also added Ref. [1] when noting the sensitivity of estimating a power law from empirical data.
>
> [1] Clauset, Shalizi, and Newman. "Power-law distributions in empirical data." SIAM Review (2009).
>
> **[Q2: Dependence on D]** The monotonic increase in matching motif distance and one-step error as the query length $D$ is increased implies that, for a given context length $L$, it becomes harder to find a "good match" in the context when the query grows too long. The reviewer is right that when the context length is sufficiently long, more delayed embedding is desirable. However, when context length is limited, maximizing the number of non-overlapping comparisons $\sim L/D$ helps. Moreover, for the one-step error in Figure 4, a short matching motif can already provide enough forecasting power to ensure high accuracy. For longer rollouts, one will need longer matching motifs to make accurate multi-step forecasts.
>
> **[Q3: Extend the context]** We can see this explicitly from Figure 10. For small context windows, the valid prediction time decreases with the embedding dimension $D$. For large context windows, this trend is reversed, and larger embedding dimension $D$ provides better forecast accuracy. This is intuitive: when context data is limited, we want to prioritize finding a match, in which case using a short motif (small $D$) is advantageous. For long context, we want to find the most informative match (coincide on many data points), in which case one should use a long motif (large $D$).
>
> Inspired by comments from other reviewers, we also added additional baselines such as Moirai, DynaMix, simplex projection, and AutoARIMA. We thank the reviewer again for their insightful comments. Please let us know if there are additional questions that we can help address.

---

> > ### Comment · Reviewer_PSB6 · 2025-11-27
> >
> > I appreciate the comprehensive reply from the authors and look forward to the updated manuscript.

---

### Official Review · Reviewer_xP7T · 2025-10-31

**Soundness:** 3
**Presentation:** 3
**Contribution:** 3
**Rating:** 4
**Confidence:** 4

**Summary:**

The paper proposes context parroting, a phenomenon observed in recent time series foundation models (FMs), as a tough-to-beat baseline for zero-shot forecasting of physical dynamical systems. The manuscript boils context parroting down to an efficient algorithm that directly copies from the context and is on par with or better than current foundations models such as Chronos. The method is used to explain recent in-context neural scaling laws and connected to the induction heads theory of LLMs. In that, context parroting highlights current failure modes of time series foundation models with the aim to improve future model design.

**Strengths:**

1. The paper is well written and structured and hence easy to follow.
2. Boiling down the algorithm behind context parroting to a lightweight routine which can be used as a baseline comparison is a great contribution to the field.
3. Using and comparing the method to existing ideas and routines from dynamical systems theory is great way to gain a more mechanistic understanding of the underlying principles of current time series FMs.

**Weaknesses:**

The paper should discuss recent work introducing DynaMix [1]. It is a FM for time series based on RNNs and MoE and shows promising performance in forecasting chaotic dynamics from limited context without context parroting. This work also addresses the finding of [2], and shows that while context parroting games short-term forecast metrics, it only produces cyclic patterns in the long-term, rendering the method unable to truly produce the chaotic dynamics underlying the system hinted by the context. An analysis that [1] includes and this paper misses, in my opinion, is one that evaluates invariant measures for long autoregressive roll-outs e.g. by estimating the empirical max. Lyapunov exponent $\lambda_{max}$ for trajectories pushing $T \rightarrow \infty$. [1] clearly shows that Chronos (which is context parroting) only produces $\lambda_{max} \approx 0$, indicating cycles (albeit high-order and complex at times). The manuscript should include a thorough discussion on these results and ideally include DynaMix as a comparison FM.

**References**:

[1] Hemmer, Christoph Jürgen, and Daniel Durstewitz. "True zero-shot inference of dynamical systems preserving long-term statistics." arXiv preprint arXiv:2505.13192 (2025).

[2] Zhang, Yuanzhao, and William Gilpin. "Zero-shot forecasting of chaotic systems." arXiv preprint arXiv:2409.15771 (2024).

**Questions:**

1. “Despite the theoretical correspondence, however, numerically it is challenging to accurately estimate the scaling coefficient α due to noise in the data.” I am confused, in case of controllable benchmark data such as datasets derived from *dysts*, how is noise a problem? Can’t we simply generate data w/o noise?
2. “Moreover, even when restricted to parroting, Chronos can in principle dynamically choose the optimal embedding dimension D for each individual time series, giving it an advantage over parroting algorithms with a fixed D.” Can the authors explain how Chronos would do this?
3. Could the authors include an experiment that shows whether the estimated max. Lyapunov exponent converges to the ground truth for increasing context lengths, where $\lambda_{max}$ is estimated from predicted trajectories only?  And of course predicting for much longer than 10 Lyapunov times, say 100-1000, to really probe the long-term climate of the forecasts of the introduced context parroting algorithm.

---

> ### Author Response · Authors · 2025-11-25
>
> We thank the reviewer for their constructive feedback. In our substantial revision, we have performed all of the suggested experiments. We hope the reviewer will find that their comments were addressed substantively:
>
> **[Weakness 1: DynaMix]** The DynaMix code was released about a month ago. It thus wasn't possible to include results on this model for the initial ICLR deadline. We have now added this baseline to Figure 2 (dysts dataset) and Tables 1-3 (SciML dataset). Both parroting and DynaMix capture the invariant proprieties of the dynamics well (much better than the other foundation models). In this regard, DynaMix is a leading model with the unique ability to capture the long-term climate of dynamical systems and generate faithful yet complex forecasts. For point forecast accuracy (sMAPE, MAE, MSE, etc.), parroting currently still outperforms DynaMix. Additional results on DynaMix can be found in Figures 6, 7, 12 and Table 4 in the Appendix. The relevant tables are attached below.
>
> **MAE @ 50 steps** for SciML tasks. **Bold = best**, *italic = second and third best*.
> | Task       | Parrot          | DynaMix        | Chronos         | Chronos Bolt    | TimesFM         | TimeMoE         | Moirai           |
> |------------|------------------|----------------|------------------|------------------|------------------|------------------|-------------------|
> | Turbulence | *0.403±0.210*    | 0.505±0.247    | 0.431±0.237      | 0.567±0.247      | 0.510±0.174      | *0.394±0.172*    | **0.382±0.189**   |
> | ECG        | **0.624±0.315**  | 0.777±0.241    | 0.873±0.422      | 0.752±0.279      | *0.723±0.259*    | 0.799±0.158      | *0.684±0.237*     |
> | Circuit    | **0.083±0.050**  | 0.425±0.172    | *0.111±0.065*    | 0.349±0.120      | *0.196±0.090*    | 0.206±0.102      | 0.213±0.093       |
> | Kuramoto   | **0.004±0.001**  | 0.076±0.002    | 0.072±0.029      | 0.961±0.084      | 0.624±0.061      | *0.070±0.011*    | **0.004±0.001**   |
>
> **MSE @ 50 steps** for SciML tasks. **Bold = best**, *italic = second and third best*.
> | Task       | Parrot           | DynaMix        | Chronos         | Chronos Bolt    | TimesFM         | TimeMoE         | Moirai           |
> |------------|-------------------|----------------|------------------|------------------|------------------|------------------|-------------------|
> | Turbulence | *0.322±0.333*     | 0.490±0.4530   | 0.380±0.408      | 0.531±0.447      | 0.403±0.262      | **0.278±0.268**  | **0.278±0.267**   |
> | ECG        | *0.916±0.630*     | 1.063±0.488    | 1.461±1.097      | 0.950±0.581      | 0.940±0.530      | *0.893±0.287*    | **0.851±0.488**   |
> | Circuit    | **0.012±0.016**   | 0.297±0.294    | *0.024±0.030*    | 0.181±0.122      | *0.065±0.056*    | 0.076±0.080      | 0.075±0.060       |
> | Kuramoto   | **0.001±0.002**   | 0.006±0.001    | 0.009±0.007      | 1.296±0.188      | 0.512±0.096      | *0.008±0.002*    | **0.001±0.001**   |
>
> **KL Divergence between predicted and true attractors** for SciML tasks. **Bold = best**, *italic = second and third best*.
> | Task       | Parrot           | DynaMix         | Chronos         | Chronos Bolt    | TimesFM         | TimeMoE         | Moirai           |
> |------------|-------------------|------------------|------------------|------------------|------------------|------------------|-------------------|
> | Turbulence | *0.028±0.044*     | **0.005±0.008**  | 0.041±0.046      | 0.048±0.058      | 0.111±0.072      | 0.070±0.058      | *0.030±0.041*     |
> | ECG        | **0.065±0.089**   | *0.099±0.104*    | 0.403±0.367      | 0.253±0.185      | 0.220±0.153      | *0.188±0.094*    | 0.276±0.311       |
> | Circuit    | *0.572±0.082*       | 2.940±0.528      | *0.630±0.118*    | 1.710±0.255      | **0.383±0.087**  | 0.816±0.200      | 0.848±0.155       |
> | Kuramoto   | **0.001±0.001**   | 1.010±0.150      | 0.537±0.087      | 3.116±0.202      | 4.489±0.363      | *0.076±0.040*    | *0.010±0.011*     |

---

> > ### Author Response · Authors · 2025-11-25
> >
> > The referee is right that, because parroting by definition produces cyclic patterns, the generated trajectory should have its maximum Lyapunov exponent equal to zero. However, we note that as the context length becomes longer, context parroting will be able to parrot increasingly more complex patterns with longer periods. As a result, the finite-time Lyapunov exponents of the generated trajectory will be nonzero and can approach the true Lyapunov exponents. We demonstrate this in the new Table 4, where parroting is able to capture the Lyapunov exponents of the chaotic systems in dysts almost as accurately as DynaMix. We attach the table below.
> >
> > | Metric                        | Parrot           | Chronos         | Dynamix          | Simplex         |
> > |------------------------------|-------------------|------------------|-------------------|------------------|
> > | Attractor KL Divergence      | **0.412 ± 0.141** | 0.679 ± 0.101    | *0.508 ± 0.147*   | 0.546 ± 0.140    |
> > | Fractal Dimension Correlation| **0.723 ± 0.042** | 0.120 ± 0.118    | *0.521 ± 0.057*   | 0.341 ± 0.072    |
> > | Largest Lyapunov Correlation | *0.343 ± 0.018*   | 0.269 ± 0.114    | **0.466 ± 0.071** | *0.343 ± 0.085*  |
> >
> > KL Divergence and correlation of invariant properties between predicted and true attractors for long forecast horizons spanning over $300$ Lyapunov times, evaluated on the dysts dataset. Error bars are standard deviation across all attractors for the KL Divergence, and uncertainty bounds based on the p-value for correlations. **Bold = best**, *italic = second best*.

---

> > > ### Author Response · Authors · 2025-11-25
> > >
> > > **[Q1: Scaling Estimator]** Thanks for pointing this out. The issue is the fractal dimension estimator (the Grassberger-Procaccia algorithm). Estimating the correlation dimension requires fitting a power law to an empirical curve showing the number of neighbors around each point as a function of distance. The problem is that power law fits are notoriously fragile to generate from finite-resolution time-series data [1]. As a result, the estimated scaling coefficient can vary significantly with granularity or hyperparameters of the power law estimator, and it will not necessarily converge with more data. This is an intrinsic problem with this calculation. However, in the DS community, correlation dimension is nonetheless considered one of the more reliable data-driven invariant quantities that can be estimated directly from time series (data-driven Lyapunov exponent estimators such as the Rosenstein algorithm are even more fragile). We've revised the paper to clarify that the numerical challenge is a limitation of the metric, not the dataset or experiments themselves. We have also added Ref. [1] when noting the sensitivity of estimating a power law from empirical data.
> > >
> > > [1] Clauset, Shalizi, and Newman. "Power-law distributions in empirical data." SIAM Review (2009).
> > >
> > > **[Q2: Selecting Optimal Embedding]** We agree this needed clarification. We have modified the text to clarify that this is a general statement: since attention is $\sim\mathcal{O}(L^2)$ operations over a context of $L$ tokens, it can perform more complex operations than nearest-neighbor search, which has time complexity $\sim\mathcal{O}(L)$.
> > >
> > > **[Q3a: Maximum Lyapunov Exponent variation with context]** We have added this experiment in the Appendix (Fig. 12). Estimating Lyapunov exponents purely from time series is highly unreliable, and we have opted to use the Rosenstein algorithm. We also include two other more stable metrics, the KL Divergence and the fractal dimension. The results confirm that context parroting, which simply samples sequences from the true measure, performs well on global metrics. We note that DynaMix also performs well, implying that it effectively leverages long context and preserves invariant properties.
> > >
> > > **[Q3b: Maximum Lyapunov Exponent variation with forecast horizon]** We have added this experiment in the Appendix (Table 4), though we note the same caveats with the Lyapunov exponent calculation as we note above for **[Q3a]**. We have thus also included KL divergence and fractal dimension as global metrics. Parroting generally performs well in this setting, but we note that DynaMix pulls ahead in terms of preserving the maximum Lyapunov exponent.
> > >
> > > We thank the reviewer again for their insightful comments. Please let us know if there are additional questions that we can help address.

---

> > > > ### Comment · Reviewer_xP7T · 2025-11-27
> > > >
> > > > **[Re: Weakness 1: DynaMix]**
> > > >
> > > > I understand that DynaMix’s code was not published in time for the authors to include the model and I am hence very happy to see that the authors now include it in their suite of comparison FMs.
> > > >
> > > > However, I find the results on the evaluation of long-term dynamics pretty buffling.
> > > >
> > > > 1) The results on the KL divergence in Figure 2 seem to contradict the results of Fig. 6 in [1]. I think this is due to the small prediction length of only 300 time steps (even less than the actual context!). Neither is this enough for the forecasters to reach their limiting dynamics, for which the KL measure was designed, nor are 300 data points in $\geq 3$ dimensions statistically sufficient to evaluate probability distributions. While Table 4 closes the gap (due to the long roll-outs), it is still puzzling that parroting achieves a better KL divergence than DynaMix despite only producing cyclic dynamics. Could it be the case that the width of the Gaussians was chosen too wide, such that differences between complex limit cycles and chaotic attractors become blurred? Can the authors graphically illustrate their choices? Or could this simply be an artifact of the presumably univariate, instead of truly multidimensional, evaluation of the KL? Did the authors assemble 3d spaces for Chronos from 3 separate univariate forecasts for each dimension, or did they perform a delay embedding?
> > > >
> > > >
> > > > 2) Similarly, the finding that KL divergence *increases* with longer context in Figure 12 contradicts with Figure 3 in [1], where long-term metrics *decrease* with context length for DynaMix. It does indeed make much more sense that performance improves with context length, as the model has more information of the underlying attractor to base its predictions on.  How do the authors explain this strange result and discrepancy? Could it be related to the issues noted above? Can the authors add more experimental details on the generation of Fig. 12? Most importantly, which prediction length was chosen?
> > > >
> > > >
> > > > 3) I find the result on Lyapunov exponents in Table 4 also contradicts Fig. 6d,e in [1]: Chronos (which is context parroting) produces $\lambda_{max} \approx 0$, while estimates based on DynaMix’s trajectories are much closer to the ground truth, which is significantly larger than 0 for chaotic systems. In Table 4 it is not clear that Chronos (and context parroting) only produce cyclic patterns (no matter how complex). If the max. Lyapunov exponent is indeed estimated from the entire predicted trajectory (9488 time points), the estimate should converge towards 0 for parroting and Chronos, which contradicts the comparably high correlation reported in Table 4. Correlations can also be high, however, if there is truly a difference in Lyapunov exponents (e.g., one being positive and the other being negative). Since the *absolute agreement* and not the correlation is important in this context, the authors should report and graph (for some examples) the distributions of actual Lyapunov exponents compared to the ground truth for Chronos, with MAE instead of correlation which can be high even when the absolute difference in exponents is large! This would be much more conclusive.
> > > >
> > > > I would appreciate if the authors would provide **a)** the Hellinger distance between power spectra [1] as an additional measure of long-term statistics, **b)** plots of power spectra that highlight the periodic nature of context parroting forecasts, **c)** address the discrepancy (see point 2)), and **d)** provide more details on the computation of the max. Lyapunov exponent as well as provide a Figure that shows histograms of estimated max. Lyapunov exponents (of course normalized by the sampling frequency of the respective system) for context parroting and Chronos across all 135 systems of the dataset used.
> > > >
> > > >
> > > > **[Re: Q1: Scaling Estimator]**
> > > >
> > > > I see, thanks for the clarification.
> > > >
> > > > **[Re: Q2: Selecting Optimal Embedding]**
> > > >
> > > > I see, this is indeed a very general statement. It would, however, be interesting to follow up on this in more detail as future work!
> > > >
> > > > **[Re: Q3a: Maximum Lyapunov Exponent variation with context]**
> > > >
> > > > Again, the same algorithm has been used to produce results in Figure 6 of [1] (see also its Appx. A.1), where Chronos, a context parroting model, produces $\lambda_{max} \approx 0$ for which matches the theoretically expected value much more closely, since Chronos is only producing periodic patterns (as also evidenced by the power spectrum). The discrepancies in these results can’t only be due to unreliability or noisiness of the Rosenstein algorithm.
> > > >
> > > > **[Re: Q3b: Maximum Lyapunov Exponent variation with forecast horizon]**
> > > >
> > > > Please see 3) above in the response above.
> > > >
> > > > **Refs:**
> > > >
> > > > [1] Hemmer, Christoph Jürgen, and Daniel Durstewitz. "True zero-shot inference of dynamical systems preserving long-term statistics." arXiv preprint arXiv:2505.13192 (2025).

---

> > > > > ### Author Response · Authors · 2025-12-03
> > > > >
> > > > > We thank the reviewer for their comments. Our revised text now includes new experiments addressing the four questions asked by the referee. The code for these experiments are available on the anonymous GitHub Repo: <https://anonymous.4open.science/r/parroting-4D26>.
> > > > >
> > > > > **a.) Long-term Hellinger spectra.** See Tables 15 and 16 in the revised main text, in which we use a context length of 2000 and a prediction horizon of 10,000 for both parroting and DynaMix. In this particular regime, DynaMix and parroting perform nearly identically. This makes sense, because parroting is able to repeat very long sequences from the ground truth. For long enough contexts, the distributional properties of the parroting predictions should approach the ground truth.
> > > > >
> > > > > **b.) Plots of power spectra.** See Figures 13 and 14 in the revised main text. The spectrum of context parroting does *not* collapse, because the sequences being parroted become quite long when the context length is long. Figure 13 shows visually that the spectra from parroting can often be closer to the ground truth than DynaMix, which is consistent with our Table 15 numerical results above. Moreover, Figure 14 shows that the power spectrum of parroting improves with longer contexts, confirming the intuition stated in (a).
> > > > >
> > > > > **c.) Longer forecast horizons.** In Fig. 12, we use a prediction length of 300; however, in response to the referee’s concern, we repeated this experiment with a prediction length of 10000. See Figure 15 of the revised manuscript. The results are consistent across the two forecast lengths.
> > > > >
> > > > > **d.) Lyapunov Exponent Histograms.** See Figure 16 and Table 16 of the revised text. In our calculation, we find that Parroting still produces estimates of largest Lyapunov exponents (LLE) closer to the ground truth values across the dataset of chaotic systems. Given the distribution, most aggregation metrics (MAE, Pearson, Spearman) will agree. Regarding the referee’s question about the Lyapunov exponent computation, see our extended discussion of (3) below.
> > > > >
> > > > > Regarding the referee’s other discussion points:
> > > > >
> > > > > **(1)** We evaluate KL divergence by combining univariate forecasts from all dimensions. We used an implementation of the KL divergence in the public `dysts` library, which scales the width of each Gaussian based on the local neighborhood around each timepoint. We don’t see a change in how this metric ranks models as we vary the prediction horizon (Fig. 12 versus Fig. 15). We also repeated this calculation while varying the width of the Gaussian kernel (ranging over 0.01, 0.1, 1.0, 10.0), and we find that the relative performance of the two models remains the same across widths (Table 17 in the revised text).
> > > > >
> > > > > **(2)** See our reply to (c) above.
> > > > >
> > > > > **(3)** We are not sure whether the DynaMix authors used the Rosenstein implementation in the Python `nolds` library or a different one, and we could not find this metric in the `metrics.py` file in the current DynaMix codebase. However, all implementations of Rosenstein contain a user-chosen integration timescale. If this is chosen too small, it won’t capture the long-term dynamics; but if this timescale is too large, then the displacement distribution saturates because the system exits the exponential growth phase. Thus longer integration timescales are not necessarily better. We are using the default integration timescale used in `nolds`. If the parroted sequence is longer than this timescale, the LLE won’t approach zero, even for a periodic sequence. Our best guess is that DynaMix sets this integration timescale to a larger number than we do. In Figure 17 of the revised text, we sweep the LLE integration timescale over two orders of magnitude, and find that the results are surprisingly stable, though DynaMix improves at longer timescales (this is consistent with our results in Table 4).
> > > > >
> > > > > From this discussion of metrics, it is clear that many long-term metrics suffer from dependence on hyperparameters: KL Divergence depends on the width of the Gaussians, LLE depends on integration timescale, and the Hellinger distance (as implemented in the DynaMix codebase) applies a smoothing kernel with a user-chosen scale.

---

> > > > > > ### Author Response · Authors · 2025-12-03
> > > > > >
> > > > > > Finally, we would like to make a few more philosophical points:
> > > > > >
> > > > > > 1. Parroted trajectories can have positive *finite-time* Lyapunov exponents, which is the only thing one can practically estimate from finite-length time series. Moreover, reconstructing the Lyapunov spectrum and the underlying attractor from a short context trajectory is an ill-defined problem to begin with (e.g., it is impossible to even determine whether the underlying dynamics are chaotic or periodic with long periods).
> > > > > > 2. Since parroting can in principle copy an arbitrarily long trajectory from the context (which represents ground truth), it can fool any algorithm that estimates Lyapunov exponents from finite-length time series. Moreover, the estimated Lyapunov exponents from a long parroted trajectory will approach the real Lyapunov exponents. In this sense, parroting is able to capture the Lyapunov spectra of any chaotic dynamics, despite only producing periodic trajectories by definition.
> > > > > > 3. The key point of our paper is that most foundation models underperform relative to the naive parroting strategy under many relevant metrics. This point is largely independent from the  technical discussions here on Lyapunov exponent estimators and prediction horizons.

---

### Meta-Review · Area_Chair_Lgjz · 2026-01-06

**Summary:**

All reviewers agreed that the paper introduces a simple yet interesting method that outperforms more complex foundation models on chaotic time-series across many settings. Reviewers also highlighted the practical usefulness of the approach to the broader research community. One review with an unusually high score of 10 (Reviewer 2L3R) appears to have been fully AI-generated and was flagged during the review process; it should therefore be disregarded. Even excluding this review, the paper received a clearly positive overall assessment. I therefore recommend acceptance.

**Reviewer Concerns:**

Only one reviewer (Reviewer xP7T) did not recommend acceptance initially. While it is difficult to determine whether this reviewer would have been fully satisfied by the rebuttal, the remaining point of discussion is largely peripheral to the paper’s core contribution and mostly concerns the computation of additional results requested by Reviewer xP7T. In light of the otherwise positive assessments from the other reviewers, it is plausible that Reviewer xP7T would have raised their score to a borderline acceptance.

**Reviewer Scores:**

See above.

---

### Decision · Program_Chairs · 2026-01-26

Accept (Poster)